# Decentralized Online Learning in General-Sum Stackelberg Games

**Yaolong Yu**[1]

**Haipeng Chen**[2]

[1]Department of Computer Science and Engineering, The Chinese University of Hong Kong, Hong Kong
[2]Data Science, William & Mary, Williamsburg, Virginia, USA

## Abstract

We study an online learning problem in general-sum Stackelberg games, where players act in a decentralized and strategic manner. We study two settings depending on the type of information for the follower: (1) the *limited information* setting where the follower only observes its own reward, and (2) the *side information* setting where the follower has extra side information about the leader's reward. We show that for the follower, myopically best responding to the leader's action is the best strategy for the limited information setting, but not necessarily so for the side information setting – the follower can manipulate the leader's reward signals with strategic actions, and hence induce the leader's strategy to converge to an equilibrium that is better off for itself. Based on these insights, we study decentralized online learning for both players in the two settings. Our main contribution is to derive last-iterate convergence and sample complexity results in both settings. Notably, we design a new manipulation strategy for the follower in the latter setting, and show that it has an intrinsic advantage against the best response strategy. Our theories are also supported by empirical results.

## 1  INTRODUCTION

Many real-world problems such as optimal auction [Myerson, 1981, Cole and Roughgarden, 2014] and security games [Tambe, 2011] can be modeled as a hierarchical game, where the two levels of players have asymmetric roles and can be partitioned into a *leader* who first takes an action and a *follower* who responds to the leader's action. This class of games is called Stackelberg or leader-follower games [Stackelberg, 1934, Sherali et al., 1983]. General-sum Stackelberg games [Roughgarden, 2010] are a class of Stackelberg games where the sum of the leader's and follower's rewards is not necessarily zero. They have broad implications in many other real-world problems such as taxation policy making [Zheng et al., 2022], automated mechanism design [Conitzer and Sandholm, 2002], reward shaping [Leibo et al., 2017], security games [Gan et al., 2018, Blum et al., 2014], anti-poaching [Fang et al., 2015], and autonomous driving [Shalev-Shwartz et al., 2016].

We focus on the setting of a pure strategy in repeated general-sum Stackelberg games follows that of Bai et al. [2021], where players act in an *online*, *decentralized*, and *strategic* manner. *Online* means that the players learn on the go as opposed to learning in batches, where regret is usually used as the learning objective. *Decentralized* means that there is no central controller, and each player acts independently. *Strategic* means that the players are self-interested and aim at maximizing their own utility. Moreover, we take a *learning* perspective, meaning that the players learn from (noisy) samples by playing the game repeatedly. A comparable framework has been investigated in numerous related studies [Kao et al., 2022], finding widespread applications in diverse real-world scenarios, such as addressing the optimal taxation problem in the AI Economist [Zheng et al., 2022] and optimizing auction procedures [Amin et al., 2013].

There have been extensive studies on decentralized learning in multi-agent games [Blum and Mansour, 2007, Wu et al., 2022, Jin et al., 2021, Meng et al., 2021, Wei et al., 2021, Song et al., 2021, Mao and Başar, 2023, Zhong et al., 2023, Ghosh, 2023], mostly focusing on settings where all agents act *simultaneously*, without a hierarchical structure. Multi-agent learning in Stackelberg games has been relatively less explored. For example, Goktas et al. [2022], Sun et al. [2023] study zero-sum games, where the sum of rewards of the two players is zero. Kao et al. [2022], Zhao et al. [2023] study cooperative games, where the leader and the follower share the same reward. Though these studies make significant contributions to understanding learning in Stackelberg games, they make limiting assumptions about the reward structures of the game. The first part of this paper studies the

learning problem in a more generalized setting of general-sum Stackelberg games. Due to the lack of knowledge of the reward structures, learning in general-sum Stackelberg games is much more challenging, and thus remains open.

Furthermore, in these studies [Goktas et al., 2022, Sun et al., 2023, Kao et al., 2022, Zhao et al., 2023], a hidden assumption is that the optimal strategy of the follower is to best respond to the leader's strategy in each round. Recent studies [Gan et al., 2019a,b, Nguyen and Xu, 2019, Birmpas et al., 2020, Chen et al., 2022, 2023] show that under information asymmetry, a strategic follower can manipulate a *commitment* leader by misreporting their payoffs, so as to induce an equilibrium different from Stackelberg equilibrium that is better-off for the follower. The second part of this paper extends this intuition to an online learning setting, where the leader learns the commitment strategy via no-regret learning algorithms (e.g., EXP3 [Auer et al., 2002b] or UCB [Auer et al., 2002a]).

We use an example online Stackelberg game to further illustrate this intuition, the (expected) payoff matrix of which is shown in Table 1. By definition, $(a_{se}, b_{se}) = (a_1, b_1)$ is its Stackleberg equilibrium. This is obtained by assuming that the follower will best respond to the leader's action. In online learning settings, when the leader uses a no-regret learning algorithm, and the (strategic) follower forms a response function as $\mathcal{F} = \{\mathcal{F}(a_1) = b_2, \mathcal{F}(a_2) = b_1\}$, then the leader will be tricked into believing that the expected payoffs of actions $a_1$ and $a_2$ are respectively 0.1 and 0.2. Hence, the leader will take action $a_2$ when using the no-regret learning algorithms, and the game will converge to $(a_2, b_1)$, where the follower has a higher payoff of 1 compared to 0.1 in the Stackleberg equilibrium.

| Leader / Follower | $b_1$ | $b_2$ |
|---|---|---|
| $a_1$ | (0.3, 0.1) | (0.1, 0.05) |
| $a_2$ | (0.2, 1) | (0.3, 0.1) |

Table 1: Payoff matrix of an example Stackelberg game. The row player is the leader. Each tuple denotes the payoffs of the leader (left) and the follower (right).

The insights gained from the above example can be generalized. In the following, we introduce two different categories of general-sum Stackelberg games that are distinguished by whether the follower has access to the information about the reward structure of the leader: (i) **limited information:** the follower only observes the reward of its own, and (ii) **side information:** the follower has extra side information of the leader's reward in addition to itself.

In the limited information setting, the follower is not able to manipulate the game without the leader's reward information. Therefore, best responding to the leader's action is indeed the best strategy. This constitutes a typical general-sum Stackelberg game. Our contribution in this setting is to prove the convergence of general-sum Stackelberg equilib-

rium when both players use (variants of) no-regret learning algorithms. Note that this is a further step after [Kao et al., 2022] who focus on cooperative Stackelberg games where the two players share the same reward. In addition, we prove last-iterate convergence [Mertikopoulos et al., 2018, Daskalakis and Panageas, 2018, Lin et al., 2020, Wu et al., 2022] results, which are considered stronger than the average convergence results.

In the side information setting, we first consider the case when the follower is omniscient, i.e., it knows both players' exact true reward functions. Building on the intuition from the above example, we design FBM, a manipulation strategy for the follower and prove that it gains an intrinsic advantage compared to the best response strategy. Then, we study a more intricate case called noisy side information, where the follower needs to learn the leader's reward information from noisy bandit feedback in the online process. We design FMUCB, a variant of FBM that finds the follower's best manipulation strategy in this case, and derive its sample complexity as well as last-iterate convergence. Our results complement existing works [Birmpas et al., 2020, Chen et al., 2022, 2023] that focus on the learning perspective of only the follower against a *commitment* leader in *offline* settings.

To validate the theoretical results, we conduct synthetic experiments for the above settings. Empirical results show that: 1) in the limited information setting, (variants of) no-regret learning algorithms lead to convergence of Stackelberg equilibrium in general-sum Stackelberg games, 2) in the side information setting, our proposed follower manipulation strategy does introduce an intrinsic reward advantage compared to best responses, both in the cases of an omniscient follower and noisy side information.

## 2 RELATED WORK

**Decentralized learning in simultaneous multi-agent games.** This line of works has broad real-world applications, e.g., when multiple self-interested teams patrol over the same targets in security domains [Jiang et al., 2013] or wildlife conservation [Ministry of WWF-Pakistan, 2015], or when different countries that independently plan their own actions in international waters against illegal fishing [Klein, 2017]. There have been extensive studies in this category [Blum and Mansour, 2007, Wu et al., 2022, Jin et al., 2021, Meng et al., 2021, Wei et al., 2021]. The seminal work of Blum and Mansour [2007] shows that decentralized no-regret learning in multi-agent general-sum games leads to a coarse correlated equilibrium (CCE). The result has been improved using more sophisticated methods like optimistic hedge [Daskalakis et al., 2021, Chen and Peng, 2020, Anagnostides et al., 2022]. Jin et al. [2021], Liu et al. [2021] study a similar question in Makov games that involves sequential decision and reinforcement learning.

**Learning in Stackelberg games.** While our work also focuses on decentralized learning, we focus on learning in Stackelberg games where the players act sequentially. Lauffer et al. [2022] study a different setting than ours where the leader first commits to a *randomized* strategy, and the follower observes the randomized strategy and best responds to it. Such a setting also appears in Balcan et al. [2015]. In our setting, the leader first plays a *deterministic* action, and the follower responds to the action after observing it, a setting that is closer to Bai et al. [2021], Kao et al. [2022]. Many past works about learning in Stackelberg games, like security games [Blum et al., 2014, Peng et al., 2019, Balcan et al., 2015, Letchford et al., 2009], only focus on learning from the leader's perspective, assuming access to an oracle of the follower's best response. More recently, there have been rising interests for Stackelberg games involving learning of both players [Goktas et al., 2022, Sun et al., 2023, Kao et al., 2022, Zhao et al., 2023]. They focus on sub-classes of games with specific reward structures. Goktas et al. [2022], Sun et al. [2023] study zero-sum stochastic Stackelberg games where the sum of rewards of the two players is zero. Kao et al. [2022], Zhao et al. [2023] study cooperative Stackelberg games where the leader and the follower share the same reward. We study general-sum Stackelberg games, a more generalized setting without assuming specific reward structures like the above. The lack of prior knowledge of the reward structures, however, makes the learning problem much more challenging. As a matter of fact, learning in repeated general-sum Stackelberg games remains an open problem. The setting in Bai et al. [2021] is closer to us, where it also learns general-sum Stackelberg equilibrium from noisy bandit feedback. But their learning process needs a central device that queries each leader-follower action pair for sufficient times, making it essentially a centralized model or offline learning. We study an online learning problem in repeated *general-sum* Stackelberg games, where the players act in a *decentralized* and *strategic* manner. Last and very importantly, a hidden assumption in these works is that the follower's best strategy is to best respond to the leader's actions. In addition to studying Stackelberg games with a best-responding follower (Section 4), we also take a new perspective of a manipulative follower (Sections 5-6), and show that manipulative strategies can indeed yield higher payoff for the follower in an online learning setting.

**Follower manipulation in Stackelberg games.** It is known that the leader has the first-mover advantage in Stackelberg games: an optimal commitment in a Stakelberg equilibrium always yields a higher (or equal) payoff for the leader than in any Nash equilibrium of the same game [Von Stengel and Zamir, 2010]. This result is under the assumption that the leader has full information about the follower's reward, or that it can learn such information via interacting with a truthful follower [Letchford et al., 2009, Blum et al., 2014, Haghtalab et al., 2016, Roth et al., 2016, Peng et al., 2019]. In other words, they assume that follower will always play

myopic best responses to the leader's strategy. Recent studies showed that a follower can actually manipulate a commitment leader, and induce an equilibrium different from the Stackelberg equilibrium by misreporting their payoffs [Gan et al., 2019a,b, Nguyen and Xu, 2019, Birmpas et al., 2020, Chen et al., 2022, 2023]. In parallel with these works, we consider an *online* learning setup where *both players* learn their best strategies from noisy bandit feedback. This is in contrast to the above existing works which take the learning perspective of only the follower against a commitment leader in an offline setup.

## 3 PRELIMINARY

A general-sum Stackelberg game is represented as a tuple $\{\mathcal{A}, \mathcal{B}, \mu_l, \mu_f\}$. $\mathcal{A}$ represents the action set of leader $l$, $\mathcal{B}$ represents the action set of follower $f$, and $|\mathcal{A}| = A$, $|\mathcal{B}| = B$. We denote $\mathcal{A} \times \mathcal{B}$ the joint action set of the leader and the follower. For any joint action $(a, b) \in \mathcal{A} \times \mathcal{B}$, we use $r_l(a, b) \in [0, 1]$ and $r_f(a, b) \in [0, 1]$ to respectively denote the noisy reward for the leader and the follower, with expectations $\mu_l(a, b) \in [0, 1]$ and $\mu_f(a, b) \in [0, 1]$. The follower has a response set to the leader's actions, $\mathcal{F} = \{\mathcal{F}(a) | a \in \mathcal{A}\}$, where $\mathcal{F}(a)$ is the response to leader's action $a$. The follower's *best response* towards $a$ is defined as the action that maximizes the follower's true reward:

$$\mathcal{F}_{br}(a) = \arg\max_{b \in \mathcal{B}} \mu_f(a, b), \qquad (1)$$

We denote $\mathcal{F}_{br} = \{\mathcal{F}_{br}(a) | a \in \mathcal{A}\}$ as the best response set. For simplicity, we assume that the best response to each action $a$ is unique and the game has a unique Stackelberg equilibrium $(a_{se}, b_{se})$,

$$a_{se} = \arg\max_{a \in \mathcal{A}} \mu_l(a, \mathcal{F}_{br}(a)), \; b_{se} = \mathcal{F}_{br}(a_{se}). \quad (2)$$

A *repeated* general-sum Stackelberg game is played iteratively in a total of $T$ rounds. In each round $t$, the procedure is as follows:

- The leader plays an action $a_t \in \mathcal{A}$.
- The follower observes the leader's action $a_t$, and plays $b_t = \mathcal{F}_t(a_t)$ as the response.
- The leader receives a noisy reward $r_{l,t}(a_t, b_t)$.
- The follower receives reward information $r_t(a_t, b_t)$ which we will elaborate more in the following.

In the last step, as motivated by the example in Table 1, we study two types of settings depending on the kind of reward information that is available to the follower:

- **Limited information.** The follower observes a (noisy) reward of its own: $r_t(a_t, b_t) = r_{f,t}(a_t, b_t)$.
- **Side information.** The follower has extra side information, meaning that it also knows the reward of the leader in addition to its own. This setting can be further divided

into the case of an **omniscient follower** who knows the exact reward functions $\mu_l$ and $\mu_f$ directly; or the case of **noisy side information** where the follower learns the rewards from noisy bandit feedback in each round, i.e., $r_t(a_t, b_t) = (r_{l,t}(a_t, b_t), r_{f,t}(a_t, b_t))$.

In the following, we will present our results for the limited information setting (Section 4), the omniscient follower setting (Section 5) and the noisy side information setting (Section 6).

# 4 A MYOPIC FOLLOWER WITH LIMITED INFORMATION

We first investigate the follower learning problem with limited information, where the best response is truly the follower's best strategy. Then, We design appropriate online learning algorithms for the leader and prove their last-iterate convergence for the entire game involving both players.

## 4.1 ALGORITHM FOR THE MYOPIC FOLLOWER

In the limited information setting, the follower does not know the entire game's payoff matrix, and therefore does not have the ability to manipulate the game. Hence, myopically learning the best response for each action $a \in \mathcal{A}$ is indeed the best strategy for the follower. The learning problem for the follower then is equivalent to $A$ independent stochastic bandit problems, one for each $a \in \mathcal{A}$ that the leader plays. Upper Confidence Bound (UCB) is a good candidate algorithm for the follower as it is asymptotically optimal in stochastic bandits problems [Bubeck et al., 2012].

In each round $t$, the follower selects its response based on UCB:
$$b_t = \arg\max_{b \in \mathcal{B}} \text{ucb}_{f,t}(a_t, b),$$

where $\text{ucb}_{f,t}(a, b) = \hat{\mu}_{f,t}(a, b) + \sqrt{\frac{2\log(T/\delta)}{n_t(a,b)}}$. $n_t(a, b) = \max\left\{1, \sum_{\tau=1}^{t-1} \mathbb{I}\{a_\tau = a \wedge b_\tau = b\}\right\}$ is the number of times that action pair $(a, b)$ has been selected, and $\hat{\mu}_{f,t}(a, b)$ is the empirical mean of the follower's reward for $(a, b)$:

$$\hat{\mu}_{f,t}(a, b) = \frac{1}{n_t(a,b)} \sum_{\tau=1}^{t-1} r_{f,\tau}(a_\tau, b_\tau) \mathbb{I}\{a_\tau = a \wedge b_\tau = b\}.$$

Following the result of [Auer et al., 2002a], for every stochastic bandit problem associated to $a \in \mathcal{A}$, UCB needs approximately $\mathcal{O}\left(\frac{B\log T}{\Delta_1^2}\right)$ rounds to find the best response, where $\Delta_1$ is the suboptimality gap to best response for all $a \in \mathcal{A}$,

$$\Delta_1 = \min_{a \in \mathcal{A}} \min_{b \neq \mathcal{F}_{br}(a)} \mu_f(a, \mathcal{F}_{br}(a)) - \mu_f(a, b).$$

This directly leads to the following proposition:

**Proposition 1.** *In a repeated general-sum Stackelberg game, if the follower uses UCB as the learning algorithm, with probability at least $1 - \delta$, the follower's regret is bounded as:*

$$\mathcal{R}_f^S(T) = \mathbb{E}\left[\sum_{t=1}^{T} \mu_f(a_t, Br(a_t)) - \mu_f(a_t, b_t)\right]$$
$$\leq \mathcal{O}\left(\frac{AB\log(T/\delta)}{\Delta_1}\right).$$

This proposition shows that a myopic follower achieves no Stackelberg regret learning when it decomposes the Stackelberg game into $A$ stochastic bandits problems and uses the UCB algorithm to respond, no matter what learning algorithm the leader uses.

It should be emphasized that the utilization of arbitrary no-regret algorithms does not inherently ensure the convergence to the Stackelberg equilibrium in the general-sum Stackelberg game. The following result is provided to support this critical observation:

**Theorem 1.** *[Non-Convergence to the SE] Applying the UCB-UCB algorithm within a repeated general-sum Stackelberg game can result in the leader suffering regret linear in $T$, which implies the game fails to converge to the Stackelberg equilibrium.*

So to further understand the convergence of the entire game, we investigate two kinds of algorithms for the leader, a UCB-style algorithm and an EXP3-style algorithm, two of the most prevalent algorithms in the online learning literature. Combining the follower's UCB algorithm, we have two decentralized learning paradigms: EXP3-UCB and UCB-UCB. It is worth noting that UCB-UCB is also studied in Kao et al. [2022] which focuses on a cooperative Stackelberg game setting. In the following, we present our first two major results on the last-iterate convergence for the two learning paradigms.

## 4.2 LAST-ITERATE CONVERGENCE OF EXP3-UCB

We next introduce a variant of EXP3 for the leader. We denote the unbiased reward estimation of the reward $r_{l,t}(a, \mathcal{F}_t(a))$ the leader receives in round $t$ as $\tilde{r}_t(a) = \frac{r_{l,t}(a, \mathcal{F}_t(a))}{\tilde{x}_t(a)} \mathbb{I}\{a_t = a\}$, where $x_t(a)$ is the weight for each action $a \in \mathcal{A}$. It is updated as $x_{t+1}(a) = \frac{\exp(y_t(a))}{\sum_{a \in \mathcal{A}} \exp(y_t(a))}$, where $y_t(a) = \sum_{\tau=1}^{t} \eta \tilde{r}_\tau(a)$. Let $\boldsymbol{x}_t = \langle x_t(a) \rangle$ be the weight vector of all actions $a \in \mathcal{A}$. It is initialized as a discrete uniform distribution: $\boldsymbol{x}_1 = [1/A, \cdots, 1/A]^\top$. In round $t$, the leader selects action $a_t$ according to the distribution

$$\tilde{\boldsymbol{x}}_t = (1 - \alpha)\boldsymbol{x}_t + \alpha[1/A, \cdots, 1/A]^\top.$$

The second term on the right-hand side enforces a random probability of taking any action in $\mathcal{A}$ to guarantee a minimum amount of exploration. As we will show in our proof of Theorem 2, the extra $\alpha$ exploration is essential to the convergence to Stackelberg equilibrium as it allows the follower to do enough exploration to find its best responses using its UCB sub-routine. In the following, we will simply use EXP3 to refer to the above variant of EXP3 with the explicit uniform exploration.

**Theorem 2.** *[Last-iterate convergence of EXP3-UCB under limited information] In a limited information setting of a repeated general-sum Stackelberg game with noisy bandit feedback, applying EXP3-UCB, with $\alpha = \mathcal{O}\left(T^{-\frac{1}{3}}\right)$, $\eta = \mathcal{O}\left(T^{-\frac{1}{3}}\right)$, with probability at least $1 - 3\delta$,*

$$\mathbb{P}\big[a_T \neq a_{se}\big] \leq \alpha +$$
$$A \exp\left(-\Delta_2 T^{\frac{2}{3}} + 2\sqrt{2A\log\frac{2}{\delta}}T^{\frac{1}{3}} + \frac{C_1 AB}{\Delta_1^2}\log\frac{T}{\delta}\log\frac{1}{\delta}\right),$$

*where $C_1$ is a constant, $\Delta_2$ is the leader's suboptimality reward gap to Stackelberg equilibrium:*

$$\Delta_2 = \min_{a \neq a_{se}} \mu_l(a_{se}, b_{se}) - \mu_l(a, \mathcal{F}_{br}(a)).$$

The proof is in Appendix A.2.

### 4.3 LAST-ITERATE CONVERGENCE OF UCBE-UCB

When the leader uses a UCB-style algorithm, to guarantee that the game converges to the Stackelberg equilibrium, it requires that the UCB term will always upper bound $\mu_l(a, \mathcal{F}_{br}(a))$ with high probability. This is not guaranteed with a vanilla UCB algorithm (Theorem 1) since the follower's response is unstabilized and not necessarily the best response before the follower's UCB algorithm converges. Therefore, we introduce a variant of UCB, called UCB with extra Exploration (UCBE), with a bonus term $S_0$ that ensures upper bounding $\mu_l(a, \mathcal{F}_{br}(a))$ with high probability. In UCBE, the leader chooses action $a_t = \arg\max_{a \in \mathcal{A}} \text{ucbe}_{l,t}(a)$ in round $t$ as follows

$$\text{ucbe}_{l,t}(a) = \hat{\mu}_{l,t}(a) + \sqrt{\frac{S_0}{n_t(a)}}, \ S_0 = \mathcal{O}\left(\frac{B}{\varepsilon^3}\log\frac{ABT}{\delta}\right). \tag{3}$$

Here $\varepsilon = \min\{\Delta_1, \Delta_2\}$, $\hat{\mu}_{l,t}(a)$ is the empirical mean of leader's action $a$: $\hat{\mu}_{l,t}(a) = \frac{1}{n_t(a)}\sum_{\tau=1}^{t-1} r_{f,\tau}(a_\tau, b_\tau)\mathbb{I}\{a_\tau = a\}$, and $n_t(a) = \max\{1, \sum_{\tau=1}^{t-1}\mathbb{I}\{a_\tau = a\}\}$.

**Theorem 3.** *[Last-iterate convergence of UCBE-UCB under limited information] In a limited information setting of*

*a repeated general-sum Stackelberg game with noisy bandit feedback, applying UCBE-UCB with $S_0 = \mathcal{O}\left(\frac{B}{\varepsilon^3}\log\frac{ABT}{\delta}\right)$, $\varepsilon = \min\{\Delta_1, \Delta_2\}$, for $T \geq \mathcal{O}\left(\frac{AS_0}{\Delta_1^2}\right)$, with probability at least $1 - 3\delta$, we have:*

$$\mathbb{P}\big(a_T \neq a_{se}\big) \leq \frac{\delta}{T}.$$

The proof is in Appendix A.3. This result shows that the game last-iterate converges to Stackelberg equilibrium using the decentralized online learning paradigm UCBE-UCB. It is worth noting that this is a further step after Kao et al. [2022] who prove *average convergence* of UCB-UCB in *cooperative* Stackelberg games.

## 5 A MANIPULATIVE AND OMNISCIENT FOLLOWER

In some real-world settings, the follower has extra side information about the leader's reward structure. As shown in the example in Table 1, a strategic follower can use that information to manipulate the leader's learning and induce the convergence of the game into a more desirable equilibrium. To start with, we study a simpler case where the follower is omniscient [Zhao et al., 2023], i.e., it knows the exact true rewards of the players. A manipulative follower aims to maximize its average reward:

$$\max_{\mathcal{F}} \frac{1}{T}\sum_{t=1}^{T} \mu_f(a_t, \mathcal{F}(a_t)), \ T \to \infty, \tag{4}$$

where the leader's action sequence $\{a_t\}_{t=1}^{T}$ is generated by the leader's no-regret learning algorithm. When the follower uses the response set $\mathcal{F}$, the leader (who only observes its own reward signals) will take actions that maximize its own reward given $\mathcal{F}$. When $\mathcal{F}$ is fixed, the learning problem for the leader reduces to a classical stochastic bandit problem, and the true reward for each action $a \in \mathcal{A}$ is $\mu_l(a, \mathcal{F}(a))$. The leader will eventually learn to take action $a' \in \arg\max_{a \in \mathcal{A}} \mu_l(a, \mathcal{F}(a))$ when using no-regret learning. This implies that the leader can be misled by the follower's manipulation.

Formulating the leader's action selection as a constraint, when $T \to \infty$ (i.e., at equilibrium), the above equation can be re-written in a more concrete form as:

$$\max_{\mathcal{F},(a',b')} \ \mu_f(a', b')$$
$$\text{s.t.} \ \ \mu_l(a', b') > \max_{a \neq a'} \mu_l(a, \mathcal{F}(a)), \ b' = \mathcal{F}(a') \tag{5}$$

For simplicity, we do not consider the tie-breaking rules for the leader. A more general formulation considering tie-breaking rules for the leader is discussed in Appendix C. For a manipulation strategy $\mathcal{F}$, it is associated with an action pair $(a', b')$. We call $\{\mathcal{F}, (a', b')\}$ a *qualified manipulation* if it satisfies the constraint in Eq.(5). We denote an optimal solution for the follower as $\mathcal{F}_{opt}$, and

$a_{fm} = \arg\max_{a \in \mathcal{A}} \mu_l(a, \mathcal{F}_{opt}(a))$, $b_{fm} = \mathcal{F}_{opt}(a_{fm})$. We assume that $(a_{fm}, b_{fm})$ is unique for simplicity. We can verify that $\{\mathcal{F}_{br}, (a_{se}, b_{se})\}$ is a qualified manipulation:

$$\mu_l(a_{se}, b_{se}) > \max_{a \neq a_{se}} \mu_l(a, \mathcal{F}_{br}(a)).$$

Because $(a_{fm}, b_{fm})$ corresponds to the optimal manipulation (thus maximal reward), we have:

**Proposition 2.** *For any general-sum Stackelberg game with reward class $\mu_l(\cdot, \cdot)$, $\mu_f(\cdot, \cdot)$, if the Stackelberg equilibrium is unique, then the reward gap between $(a_{fm}, b_{fm})$ and $(a_{se}, b_{se})$ satisfies:*

$$Gap(a_{fm}, a_{se}) = \mu_f(a_{fm}, b_{fm}) - \mu_f(a_{se}, b_{se}) \geq 0. \quad (6)$$

*The second equality holds if and only if $(a_{se}, b_{se}) = (a_{fm}, b_{fm})$.*

### 5.1 FOLLOWER'S BEST MANIPULATION (FBM)

---
**Algorithm 1:** Follower's best manipulation (FBM)

---
**Input:** Candidate set $\mathcal{K} = \mathcal{A} \times \mathcal{B}$, $\mu_l(\cdot, \cdot)$, $\mu_f(\cdot, \cdot)$
1: Candidate manipulation pair
   $(a', b') = \arg\max_{(a,b) \in \mathcal{K}} \mu_f(a, b)$
2: $\mathcal{F} = \{\mathcal{F}(a') = b'\} \cup \{\mathcal{F}(a) = \text{wr}(a) = \arg\min_{b \in \mathcal{B}} \mu_l(a, b) : a \neq a'\}$
3: **if** $\max_{a \neq a'} \mu_l(a, \mathcal{F}(a)) \geq \mu_l(a', b')$ **then**
4:     Eliminate $(a', b')$ from candidate set:
       $\mathcal{K} \leftarrow \mathcal{K} \backslash (a', b')$
5:     Return to Line 1
6: **end if**
**Output:** The response function $\mathcal{F}_{opt} = \mathcal{F}$

---

Based on the insights from the example in Table 1 and Proposition 2, we propose Follower's Best Manipulation (FBM, Algorithm 1), a greedy algorithm that finds the best manipulation strategy for an omniscient follower. On the high level, the key idea is to find a response set $\mathcal{F}$ that exaggerates the difference in the leader's reward when using $a'$ (an action that leads to a large follower's reward) compared to other actions (Line 2).

$\mathcal{K}$ is the candidate manipulation pair set which is initialized as the entire action space $\mathcal{A} \times \mathcal{B}$. It starts by selecting the potential manipulation pair $(a', b')$ from the candidate set $\mathcal{K}$ which maximizes $u_f(\cdot, \cdot)$ (Line 1). It then forms the manipulation strategy (response set) that returns minimum leader's reward $\min_{b \in \mathcal{B}} \mu_l(a, b)$ for $a \neq a'$, and maximum leader's reward $\mu_l(a', b')$ when plays $a'$ (Line 2). Here with a bit abuse of notation, $\text{wr}(a)$ stands for the follower's *worst response* that induces the lowest *leader*'s reward

$$\text{wr}(a) = \arg\min_{b \in \mathcal{B}} \mu_l(a, b).$$

Next (Line 3), it verifies if the current $\{\mathcal{F}, (a', b')\}$ is a qualified manipulation. If it is, then $\mathcal{F}$ is the optimal manipulation strategy since the associated manipulation pair leads to maximum reward for the follower. Hence, it exits the loop and returns the final solution $\mathcal{F}_{opt} = \mathcal{F}$. Otherwise, the algorithm eliminates $(a', b')$ from $\mathcal{K}$ and repeats the process (Lines 4-5).

Under the follower's best manipulation strategy, the leader's problem reduces to a stochastic bandit problem, where the reward for each action $a \in \mathcal{A}$ is $\mu_l(a, \mathcal{F}_{opt}(a))$, and the suboptimality gap is

$$\Delta_3 = \min_{a \neq a_{fm}} \mu_l(a_{fm}, b_{fm}) - \mu_l(a, \text{wr}(a)).$$

**Proposition 3.** *In a repeated general-sum Stackelberg game with a unique Stackelberg equilibrium, if the leader uses a no-regret learning algorithm $\mathcal{C}$ and the follower uses the best manipulation $\mathcal{F}_{opt}$,*

$$\frac{1}{T} \sum_{t=1}^{T} \mu_f(a_t, \mathcal{F}_{opt}(a_t))$$
$$= \frac{1}{T} \sum_{t=1}^{T} \mu_f(\bar{a}_t, \mathcal{F}_{br}(\bar{a}_t)) + Gap(a_{fm}, a_{se}) + \delta(T),$$

*where $\delta(T) \to 0$ as $T \to \infty$, $\{a_t\}_{t=1}^{T}$ is generated by Algorithm $\mathcal{C}$ and $\mathcal{F}_{opt}$, and $\{\bar{a}_t\}_{t=1}^{T}$ is generated by Algorithm $\mathcal{C}$ and $\mathcal{F}_{br}$.*

When the follower keeps using the best manipulation strategy, the game will eventually converge to the best manipulation pair $(a_{fm}, b_{fm})$. Similarly, if the follower uses the best response strategy, the game will eventually converge to the Stackelberg equilibrium $(a_{se}, b_{se})$. According to Propositions 2 and 3, the best manipulation strategy gains an extra average reward $Gap(a_{fm}, a_{se})$ compared to that of the best response strategy.

## 6 MANIPULATIVE FOLLOWER WITH NOISY SIDE INFORMATION

### 6.1 FOLLOWER'S MANIPULATION STRATEGY

We now study the more intricate setting of learning the best manipulation strategy with noisy side information. We first present the follower manipulation algorithm FMUCB in Algorithm 2, a variant of FBM (Algorithm 1) that uses UCB to learn the best follower manipulation strategy with noisy side information. We can see that Algorithm 2 largely resembles Algorithm 1, and hence we omit the detailed description. Except for the inherited idea from FBM to manipulate the learning of the leader, another key intuition of FMUCB is to design appropriate UCB terms in place of the true reward terms $\mu_l(a, b)$ and $\mu_f(a, b)$ in FBM, that balances the trade-off between *exploration* and *manipulation*.

**Algorithm 2:** FMUCB($t$): Follower's manipulation by UCB at round $t$

**Input:** Candidate set $\mathcal{K} = \mathcal{A} \times \mathcal{B}$, $\mathrm{U}_{l,t}(\cdot, \cdot)$, $\mathrm{U}_{f,t}(\cdot, \cdot)$, $\hat{\mu}_{l,t}(\cdot, \cdot)$.
1: Candidate manipulation pair
   $(a', b') = \arg\max_{(a,b) \in \mathcal{K}} \mathrm{U}_{f,t}(a, b)$
2: $\mathcal{F} = \{\mathcal{F}(a') = b'\} \cup \{\mathcal{F}(a) = \arg\min_{b \in \mathcal{B}} \mathrm{U}_{l,t}(a, b) :$
   $a \neq a'\}$
3: **if** $\max_{a \neq a'} \mathrm{U}_{l,t}(a, \mathcal{F}(a)) \geq \hat{\mu}_{l,t}(a', b')$ **then**
4:    Eliminate $(a', b')$ from candidate set:
     $\mathcal{K} \leftarrow \mathcal{K} \backslash (a', b')$
5:    Return to Line 1
6: **end if**
**Output:** The response function $\mathcal{F}$

In Line 1, the goal is to find the candidate manipulation pair that maximizes $u_f(\cdot, \cdot)$. Therefore, we maximize the upper confidence bound $\mathrm{U}_{f,t}(a, b)$:

$$\mathrm{U}_{f,t}(a, b) = \hat{\mu}_{f,t}(a, b) + \sqrt{\frac{2 \log(ABT/\delta)}{n_t(a, b)}}. \quad (7)$$

The suboptimal manipulation pair will not be selected after enough exploration, given the existence of the following suboptimality gap: for any action pair $(a_k, b_k) \in \mathcal{A} \times \mathcal{B}$, satisfying $\mu_f(a_k, b_k) < \mu_f(a_{fm}, b_{fm})$,

$$\Delta_4 = \min_{(a_k, b_k)} \mu_f(a_{fm}, b_{fm}) - \mu_f(a_k, b_k).$$

Conversely, in Line 2, the goal is to find the "worst response" $\mathrm{wr}(a) = \arg\min_{b \in \mathcal{B}} \mu_l(a, b)$ for each action $a \neq a_{fm}$. Therefore, we define the term $\mathrm{U}_{l,t}(a, b)$ to be the lower confidence bound:

$$\mathrm{U}_{l,t}(a, b) = \hat{\mu}_{l,t}(a, b) - \sqrt{\frac{2 \log(ABT/\delta)}{n_t(a, b)}}. \quad (8)$$

For each action $a$, the suboptimality gap against $\mathrm{wr}(a)$ is defined as $\Delta_{5,a} = \min_{b \neq \mathrm{wr}(a)} \mu_l(a, b) - \mu_l(a, \mathrm{wr}(a))$, and the minimum suboptimality gap for all $a \in \mathcal{A}$ is

$$\Delta_5 = \min_{a \in \mathcal{A}} \Delta_{5,a}.$$

Given $\Delta_5$, the suboptimal $\mathrm{wr}(a)$ will not be selected after enough exploration.

When verifying whether the current $\{\mathcal{F}, (a', b')\}$ is a qualified manipulation (Line 3), we also use $U_{l,t}(a, \mathcal{F}(a))$ as it lower bounds $\mu_l(a, \mathcal{F}(a))$. This guarantees that the true best manipulation solution satisfies the constraint since the following holds with high probability: for any action $a \neq a_{fm}$,

$$U_{l,t}(a, \mathrm{wr}(a)) \leq \mu_l(a, \mathrm{wr}(a)) < \mu_l(a_{fm}, b_{fm}).$$

This eliminates the unfeasible solution since $U_{l,t}(a, \mathrm{wr}(a))$ approaches $\mu_l(a, \mathrm{wr}(a))$, and the following suboptimality

gap exists: for any action pair $(a_k, b_k) \in \mathcal{A} \times \mathcal{B}$ satisfying $\mu_f(a_{fm}, b_{fm}) < \mu_f(a_k, b_k)$,

$$\Delta_6 = \min_{(a_k, b_k)} \max_{a \neq a_k} \mu_l(a, \mathrm{wr}(a)) - \mu_l(a_k, b_k).$$

Hence, Algorithm 2 can find the follower's best manipulation with noisy side information.

## 6.2 LAST ITERATE CONVERGENCE ANALYSIS

We also provide theoretical guarantees for the sample efficiency and convergence of this algorithm. As a continuation of Section 4, we analyze the performance of our designed algorithm towards the leader's two kinds of algorithms – EXP3 and UCBE.

**Theorem 4** (Last iterate convergence of EXP3-FMUCB with noisy side information). *When the leader uses EXP3 with parameter $\alpha = \mathcal{O}\left(T^{-\frac{1}{3}}\right)$, $\eta = \mathcal{O}\left(T^{-\frac{1}{3}}\right)$, the follower uses FMUCB, and let $T_{f,w}$ be the total number of rounds that the follower did not play the best manipulation strategy in $T$ rounds, with probability at least $1 - 3\delta$,*

$$T_{f,w} \leq \mathcal{O}\left(\frac{A^2 B}{\alpha \varepsilon^2} \log \frac{ABT}{\delta} \log \frac{1}{\delta}\right), \text{and}$$

$$\mathbb{P}[a_T \neq a_{fm}] \leq \alpha +$$

$$A \exp\left(-\Delta_3 T^{\frac{2}{3}} + 2\sqrt{2A \log \frac{2}{\delta}} T^{\frac{1}{3}} + \frac{C_2 A^2 B}{\varepsilon^2} \log \frac{ABT}{\delta} \log \frac{1}{\delta}\right),$$

*where $\varepsilon = \min\{\Delta_4, \Delta_5, \Delta_6\}$, $C_2$ is a constant.*

**Theorem 5** (Last iterate convergence of UCBE-FMUCB with noisy side information). *When the leader uses UCBE with $S_0 = \mathcal{O}\left(\frac{B}{\varepsilon^3} \log \frac{ABT}{\delta}\right)$ and $\varepsilon = \min\{\Delta_4, \Delta_5, \Delta_6\}$, the follower uses FMUCB, and let $T_{f,w}$ be the total number of rounds that the follower did not play the best manipulation strategy in $T$ rounds, with probability at least $1 - 3\delta$,*

$$T_{f,w} \leq \mathcal{O}\left(\frac{AB}{\varepsilon^2} \log \frac{ABT}{\delta}\right),$$

*and for $T \geq \mathcal{O}\left(\frac{AS_0}{\Delta_3^2}\right)$, $\mathbb{P}[a_T \neq a_{fm}] \leq \frac{\delta}{T}$.*

The proofs for Theorems 4 and 5 are respectively in Appendix B.1 and Appendix B.2. The two theorems present the sample complexity for learning the best manipulation strategy using FMUCB with noisy side information. They also guarantee that the game will achieve the last iterative convergence to the best manipulation pair $(a_{fm}, b_{fm})$ with the algorithms EXP3-FMUCB and UCBE-FMUCB.

## 7 EMPIRICAL RESULTS

To validate the theoretical results, we conduct experiments on a synthetic Stackelberg game. For both players, the number of actions $A = B = 5$. The players' mean rewards

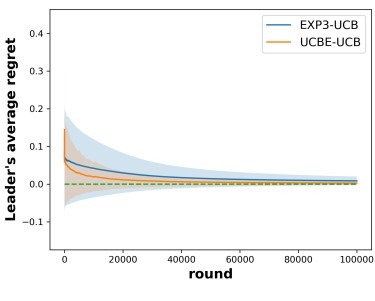
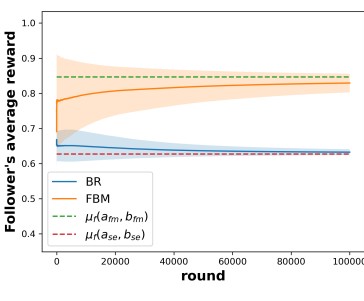
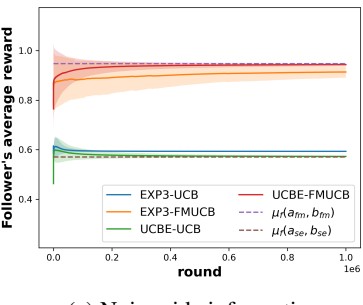

| (a) Limited information | (b) Omniscient follower | (c) Noisy side information |

Figure 1: Experiments for the limited information and side information settings. Each experiment is run 5 times to calculate the mean and std (standard deviation). The shaded area shows ±std.

$\mu_l(a, b)$ and $\mu_f(a, b)$ of each action pair $(a, b)$ are generated independently and identically by sampling from a uniform distribution $U(0, 1)$. For any action pair $(a, b)$, the noisy reward in round $t$ is generated from a Bernoulli distribution: $r_{l,t}(a, b) \sim \text{Ber}(\mu_l(a, b))$ and $r_{f,t}(a, b) \sim \text{Ber}(\mu_f(a, b))$, respectively. All experiments were run on a MacBook Pro laptop with a 1.4 GHz quad-core Intel Core i5 processor, a 1536 MB Intel Iris Plus Graphics 645, and 8 GB memory.

Our first experiment validates the convergence of no-regret learning algorithms such as EXP3-UCB and UCBE-UCB (Theorems 2 and 3 in Section 4). Figure 1a shows the average regret of the leader w.r.t. round $t$. The blue and orange curves are respectively for EXP3-UCB ($\alpha = 0.01$, $\eta = 0.001$) and UCBE-UCB. We can see that the leader achieves no Stackelberg regret learning via both algorithms, implying that the game converges to Stackelberg equilibrium in both cases.

The second experiment validates the intrinsic reward gap between FBM and BR when the follower is omniscient (Proposition 3 in Section 5). In this experiment, the leader uses EXP3 ($\alpha = 0.01$, $\eta = 0.001$), and the follower uses BR (blue curve) or FBM (red curve). The result is in Figure 1b, where the x-axis is the number of rounds $t$, and the y-axis is the average follower reward. It shows that after playing sufficient rounds, the follower's reward does converge, and FBM yields a significant intrinsic advantage (the gap between the two dashed lines) of approximately 0.22 compared to that of BR.

The third experiment validates the intrinsic reward gap between FMUCB and UCB when the follower learns the leader's reward structure via noisy bandit feedback (Theorems 4 and 5 in Section 6). The leader uses either EXP3 ($\alpha = 0.1$, $\eta = 0.001$) or UCBE, and we compare the average follower reward w.r.t. round $t$ when it uses FMUCB v.s. the baseline best response learned by a vanilla UCB. This introduces four curves, EXP3-UCB, EXP3-FMUCB, UCBE-UCB, and UCBE-FMUCB. We can see that no matter whether the leader uses EXP3 or UCBE, FMUCB yields a significant reward advantage of about 0.3 for the follower

compared to that of UCB.

## 8 CONCLUSION

We study decentralized online learning in general-sum Stackelberg games under the *limited information* and *side information* settings. For both settings, we design respective online learning algorithms for both players to learn from noisy bandit feedback, and prove the last iterate convergence as well as derive sample complexity for our designed algorithms. Our theoretical results also show that the follower can indeed gain an advantage by using a manipulative strategy against the best response strategy, a result that complements existing works in offline settings. Our empirical results are consistent with our theoretical findings.

## 9 ACKNOWLEDGMENT

The authors would like to thank Haifeng Xu and Mengxiao Zhang for valuable discussions and feedback.

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

# Decentralized Online Learning in General-Sum Stackelberg Games
## (Supplementary Material)

**Yaolong Yu**[1]                    **Haipeng Chen**[2]

[1]Department of Computer Science and Engineering, The Chinese University of Hong Kong, Hong Kong
[2]Data Science, William & Mary, Williamsburg, Virginia, USA

# A  PROOFS FOR SECTION 4

## A.1  PROOF FOR THEOREM 1

| Leader / Follower | $b_1$ | $b_2$ |
|:---:|:---:|:---:|
| $a_1$ | (0.95, 0.3) | (0.9, 0.2) |
| $a_2$ | (1, 0.8) | (0, 0.79) |

Table 2: Payoff matrix of an specific Stackelberg game.

We apply the UCB-UCB algorithm to a specific Stackelberg game, as depicted in Table 2, where the Stackelberg equilibrium is identified as $(a_2, b_1)$. The UCB algorithms for the leader and follower are defined respectively as:

$$\text{ucb}_{l,t}(a) = \hat{\mu}_{l,t}(a) + \sqrt{\frac{2\log(T/\delta)}{n_t(a)}},$$

and

$$\text{ucb}_{f,t}(a, b) = \hat{\mu}_{f,t}(a, b) + \sqrt{\frac{2\log(T/\delta)}{n_t(a, b)}}.$$

For simplicity in demonstrating the proof, it is assumed that players receive their actual true rewards at each round. Thus, $\text{ucb}_{l,t}(a_1) > 0.9$, and for $a_2$ and $b_1$, $b_2$, the follower's UCB values are calculated as:

$$\text{ucb}_{f,t}(a_2, b_1) = \mu_f(a_2, b_1) + \sqrt{\frac{2\log(T/\delta)}{n_t(a_2, b_1)}},$$

$$\text{ucb}_{f,t}(a_2, b_2) = \mu_f(a_2, b_2) + \sqrt{\frac{2\log(T/\delta)}{n_t(a_2, b_2)}}.$$

Let $n_t(a_2, b_1) = 2k_1 \log(T/\delta) + 1$ and $n_t(a_2, b_2) = 2k_2 \log(T/\delta)$. The UCB mechanism implies that for the follower to prefer $b_1$ over $b_2$, the following inequality must hold:

$$\mu_f(a_2, b_1) + \sqrt{\frac{2\log(T/\delta)}{n_t(a_2, b_1) - 1}} > \mu_f(a_2, b_2) + \sqrt{\frac{2\log(T/\delta)}{n_t(a_2, b_2)}}.$$

Given $\mu_f(a_2, b_1) - \mu_f(a_2, b_2) = 0.01$, it follows that:

$$k_2 > \frac{1}{\left(0.01 + \sqrt{\frac{1}{k_1}}\right)^2}.$$

Assuming $k_2 = 10^4$ yields $k_1 > \frac{1}{4}k_2$. If $k_1 + k_2 = \frac{5}{4} \times 10^4$, then $k_1 < 4k_2$. Given $n_t(a_2) = n_t(a_2, b_1) + n_t(a_2, b_2)$ and assuming $n_t(a_2) = \frac{5}{4} \times 10^4 \log(T/\delta)$, for sufficiently large $T$, the leader's UCB for $a_2$ becomes:

$$\text{ucb}_{l,t}(a_2) = \frac{n_t(a_2, b_1)\mu_l(a_2, b_1) + n_t(a_2, b_2)\mu_l(a_2, b_2)}{n_t(a_2)} + \sqrt{\frac{2\log(T/\delta)}{n_t(a_2)}} < 0.9,$$

which leads to a contradiction. Thus, $n_t(a_2) < \frac{5}{4} \times 10^4 \log(T/\delta) = \mathcal{O}(\log(T/\delta))$, indicating that the leader will play the optimal action no more than $\mathcal{O}(\log(T/\delta))$ times. Consequently, for sufficiently large $T$, the leader is expected to incur regret linear in $T$.

## A.2  PROOF FOR THEOREM 2

**Theorem** (Last iterate convergence of EXP3-UCB under limited information). *In a limited information setting of a repeated general-sum Stackelberg game with noisy bandit feedback, applying EXP3-UCB, with $\alpha = \mathcal{O}\left(T^{-\frac{1}{3}}\right)$, $\eta = \mathcal{O}\left(T^{-\frac{1}{3}}\right)$, for $a \neq a_{se}$, with probability at least $1 - 3\delta$,*

$$\mathbb{P}\big[a \neq a_{se}\big] \leq \alpha + A \exp\left(-\Delta_2 T^{\frac{2}{3}} + 2\sqrt{2A \log \frac{2}{\delta}} T^{\frac{1}{3}} + \frac{C_1 AB}{\Delta_1^2} \log \frac{T}{\delta} \log \frac{1}{\delta}\right),$$

*where $C_1$ is a constant, $\Delta_2$ is the leader's suboptimality reward gap to Stackelberg equilibrium:*

$$\Delta_2 = \min_{a \neq a_{se}} \mu_l(a_{se}, b_{se}) - \mu_l(a, \mathcal{F}_{br}(a)).$$

We first prove the following two lemmas which will then be used to prove Theorem 2. We denote $\tilde{r}_t(a) = \frac{1}{\overline{x}_t(a)} r_{l,t}(a, \mathcal{F}_t(a)) \mathbb{I}\{a_t = a\}$. We denote $r_t(a) = \mu_l(a, \mathcal{F}_t(a))$ and $\mu(a) = \mu_l(a, \mathcal{F}_{br}(a))$ in this section when there is no ambiguity.

**Lemma 1.** *[Wu et al., 2022] For $a \in \mathcal{A}$, $T > 0$, with probability at least $1 - \delta$,*

$$\left|\sum_{t=1}^T \eta(\tilde{r}_t(a) - r_t(a))\right| < \sqrt{2\eta^2 \frac{A}{\alpha} T \log \frac{2}{\delta}}.$$

***Proof of Lemma 1.*** Fix $T > 0$ and any action $a$. Let $\Xi_t = \eta(\tilde{r}_t(a) - r_t(a))$ and $S_t = \sum_{\tau=1}^t \Xi_\tau$. Since $T$ is fixed, $\mathbb{E}[S_t]$ is bounded. Moreover, we have

$$\mathbb{E}[S_t | S_{t-1}, \cdots, S_1] = S_{t-1} + \eta \cdot \mathbb{E}[\tilde{r}_t(a) - r_t(a) | \mathcal{G}_{t-1}] = S_{t-1}.$$

By definition, $\{S_t\}_{t=1}^T$ is a martingale. Apply Azuma's inequality to $\{S_t\}$, for any $x > 0$, $1 \leq t \leq T$, we have

$$\mathbb{P}[|S_t| \geq x] \leq 2 \exp\left(-\frac{x^2}{2W}\right). \tag{9}$$

$W$ is an upper bound of $\sum_{\tau=1}^t \mathbb{E}[\Xi_\tau^2 | \mathcal{G}_{\tau-1}]$. Note that

$$
\begin{aligned}
\mathbb{E}[\Xi_t^2 | \mathcal{G}_{t-1}] &= \eta^2 \mathbb{E}\left[(0 - r_t(a))^2 \cdot (1 - x_t(a)) + (\tilde{r}_t(a) - r_t(a))^2 \cdot x_t(a) \Big| \mathcal{G}_{t-1}\right] \\
&= \eta^2 \mathbb{E}\left[r_{l,t}^2(a, \mathcal{F}_t(a)) \cdot \frac{1}{x_t(a)} - r_t^2(a) \Big| \mathcal{G}_{t-1}\right] \\
&\leq \frac{A}{\alpha} \eta^2.
\end{aligned}
$$

We take $t = T$ and $W = \frac{A}{\alpha} \eta^2 T$ in Eq.(9) and obtain

$$x \geq \sqrt{2W \log \frac{2}{\delta}},$$

which implies with probability at least $1 - \delta$,

$$\left|\sum_{t=1}^T \eta(\tilde{r}_t(a) - r_t(a))\right| < \sqrt{2\eta^2 \frac{A}{\alpha} T \log \frac{2}{\delta}}.$$

$\square$

**Lemma 2.** *We set $\delta = T^{-\beta}$ in $ucb_{f,t}(\cdot, \cdot)$, and $\beta \geq 3$. Denote $T_{wbr}(a)$ as the number of rounds that the follower did not play best response when leader played $a$, with probability at least $1 - \delta$, for all $a \in \mathcal{A}$, we have*

$$T_{wbr}(a) \leq \mathcal{O}\left(B\frac{\log(T/\delta)}{\Delta_1^2}\right).$$

*Similarly, we denote the number of wrong recognition rounds of $wr(a)$ in Algorithm 2 for every action $a \in \mathcal{A}$ as $T_{wwr}(a)$, with probability at least $1 - \delta$, for all $a \in \mathcal{A}$, we have*

$$T_{wwr}(a) \leq \mathcal{O}\left(B\frac{\log(ABT/\delta)}{\Delta_1^2}\right).$$

*Proof.* Combining the Theorem 10.14 of Orabona [2019] and Bernstein inequality we can get the results. □

***Proof of Theorem 2.*** We denote $\mu(a) = \mu_l(a, \mathcal{F}_{br}(a))$ in this section when there is no ambiguity. For $a \neq a_{se}$, based on E.4 of Yu et al. [2022], we have

$$\sum_{t=1}^{T} r_t(a) - r_t(a_{se}) = \sum_{t=1}^{T} \left(r_t(a) - \mu(a)\right) + \left(\mu(a) - \mu(a_{se})\right) + \left(\mu(a_{se}) - r_t(a_{se})\right)$$

$$\leq \sum_{t=1}^{T} \left(r_t(a) - \mu(a)\right) - \Delta_2 T + \sum_{t=1}^{T} \left(\mu(a_{se}) - r_t(a_{se})\right) \tag{10}$$

$$\leq 4s_m \sum_{a \in \mathcal{A}} T_{wbr}(a) - \Delta_2 T,$$

where $s_m$ represents the biggest interval the leader chooses action $a$ between the $i$-th time and the $(i+1)$-th time for any $i \in [n(a) - 1]$. With probability at least $1 - \delta$, we have

$$s_m \leq \mathcal{O}\left(\frac{A}{\alpha} \log \frac{1}{\delta}\right).$$

We denote $y_t(a) = \sum_{\tau=1}^{t} \eta \tilde{r}_t(a)$. For $a \neq a_{se}$, combined with 1 and Eq.(10), we have

$$y_T(a) - y_T(a_{se}) = \sum_{t=1}^{T} \eta\left[\tilde{r}_t(a) - \tilde{r}_t(a_{se})\right]$$

$$= \sum_{t=1}^{T} \eta\left[\tilde{r}_t(a) - r_t(a)\right] + \sum_{t=1}^{T} \eta\left[r_t(a) - r_t(a_{se})\right] + \sum_{t=1}^{T} \eta\left[r_t(a) - \tilde{r}_t(a_{se})\right]$$

$$\leq 2\sqrt{2\eta^2\frac{A}{\alpha}T\log\frac{2}{\delta}} + 4\eta s_m \sum_{a \in \mathcal{A}} T_{wbr}(a) - \Delta_2 \eta T.$$

For any $a \neq a_{se}$, we set $\alpha = \mathcal{O}\left(T^{-\frac{1}{3}}\right)$, $\eta = \mathcal{O}\left(T^{-\frac{1}{3}}\right)$, by the union bound, with probability at least $1 - 3\delta$,

$$x_{T+1}(a) = \frac{\exp(y_T(a))}{\sum_{a' \in \mathcal{A}} \exp(y_T(a'))} \leq \frac{\exp(y_T(a))}{\exp(y_T(a_{se}))}$$

$$\leq \exp\left(-\Delta_2 T^{\frac{2}{3}} + 2\sqrt{2A\log\frac{2}{\delta}}T^{\frac{1}{3}} + \frac{C_1 AB}{\Delta_1^2}\log\frac{T}{\delta}\log\frac{1}{\delta}\right) \tag{11}$$

Note that

$$\mathbb{P}\left[a_T \neq a_{se}\right] = \sum_{a \neq a_{se}} \tilde{x}_T(a) \leq \alpha + \sum_{a \neq a_{se}} x_T(a).$$

So, combined with Eq.(11), we have reached our conclusion.

□

**Remark:** Last-iterate convergence refers to the phenomenon where the sequence of actions converges to the optimal action or the sequence of policies converges to the optimal policy, formulated as $\lim_{t\to\infty} a_t = a_\star$. It represents a particularly strong form of convergence, which is inevitably a stronger notion than the average-iterate convergence [Wu et al., 2022, Abe et al., 2022, Cai et al., 2024]: an algorithm demonstrating last-iterate convergence is proven to also be capable of achieving average-iterate convergence whereas the reverse is not necessarily true. Our findings corroborate this view, indicating that $\lim_{t\to\infty} P(a_t \neq a_\star) = 0$ leads to $\lim_{t\to\infty} a_t = a_\star$.

The formulations of the bounds presented in both Theorem 2 and Theorem 3 are conventional within the learning theory community, attributable to divergent analytical approaches applied to EXP3 and UCB type algorithms. Generally speaking, Theorem 2 introduces a decay rate of $\mathcal{O}(T^{-1/3} + \exp(-T^{2/3}))$, contrasting with the linear decay in $T$ observed in Theorem 3. This distinction suggests a comparative advantage for Theorem 3, contingent upon a specific constraint on $T$. Nevertheless, given that both bounds are influenced by the gaps, denoted as $\Delta_1, \Delta_2$, a thorough comparison necessitates consideration of these gap values.

## A.3 PROOF FOR THEOREM 3

**Theorem** (Last iterate convergence of UCBE-UCB under limited information). *In a limited information setting of a repeated general-sum Stackelberg game with noisy bandit feedback, applying UCBE-UCB with $S_0 = \mathcal{O}\left(\frac{B}{\varepsilon^3}\log\frac{ABT}{\delta}\right)$, $\varepsilon = \min\{\Delta_1, \Delta_2\}$, for $T \geq \mathcal{O}\left(\frac{AS_0}{\Delta_1^2}\right)$, with probability at least $1 - 3\delta$, we have:*

$$\mathbb{P}\left(a_T \neq a_{se}\right) \leq \frac{\delta}{T}.$$

**Lemma 3.** *Let $U_l(a, b) = \hat{\mu}(a, b) + \sqrt{\frac{S_1}{n_t(a,b)}}$, $n(a, b) \geq 1$, set $S_1 = 2\log\frac{ABT}{\delta}$, when $n(a, b) \geq \mathcal{O}\left(\frac{\log(ABT/\delta)}{\varepsilon^2}\right)$, with probability at least $1 - \frac{\delta}{ABT}$, we have*

$$U_l(a, b) \in \left[\mu(a, b) - \frac{\varepsilon}{4}, \mu(a, b) + \frac{\varepsilon}{4}\right].$$

*A similar result holds for the follower.*

***Proof of Lemma 3.*** (1) When $n(a, b) \geq \mathcal{O}\left(\frac{\log(ABT/\delta)}{\varepsilon^2}\right)$, by the Hoeffding inequality, with probability at least $1 - \frac{\delta}{ABT}$, we have

$$\hat{\mu}(a, b) \in \left[\mu(a, b) - \frac{\varepsilon}{8}, \mu(a, b) + \frac{\varepsilon}{8}\right].$$

(2) When $n(a, b) \geq \mathcal{O}\left(\frac{S_1}{\varepsilon^2}\right)$,

$$\sqrt{\frac{S_1}{n_t(a,b)}} \leq \frac{\varepsilon}{8}.$$

So combining (1) and (2), when $n(a, b) \geq \mathcal{O}\left(\max\left\{\frac{\log(ABT/\delta)}{\varepsilon^2}, \frac{S_1}{\varepsilon^2}\right\}\right)$, with probability at least $1 - \frac{\delta}{ABT}$, we have

$$U_l(a, b) \in \left[\mu(a, b) - \frac{\varepsilon}{4}, \mu(a, b) + \frac{\varepsilon}{4}\right].$$

The bound for the UCB term of the follower is completely analogous and thus omitted. $\qquad\square$

***Proof of Theorem 3.*** We next only consider rounds that the leader plays $a \in \mathcal{A}$. With a little abuse of notation, $t$ is referred to as $t$-th time that the leader plays the action $a$. We denote $\text{ucb}_l(a) = \hat{\mu}(a) + \sqrt{\frac{S_0}{n(a)}}$. We ignore the subscript $t$ when there is no ambiguity.

Noticed that when $n(a) < \frac{S_0}{4}$, $\text{ucb}_l(a) > 2$. And when $n(a) \geq S_0$, $\text{ucb}_l(a) \leq 2$. So when total round $T$ ($T \geq \mathcal{O}(AS_0)$) is big enough, the leader will play action $a$ for at least $\frac{S_0}{4}$ rounds.

We next bound the difference between accumulative reward the leader receives and the accumulative reward under best responses,

$$\left|\sum_{t=1}^{n(a)} r_{l,t}(a, b_t) - \mu_l(a, \mathcal{F}_{br}(a))\right|$$

$$= \left|\sum_{t=1}^{n(a)}\left(r_{l,t}(a, b_t) - \mu_l(a, \mathcal{F}_{br}(a))\right)\mathbb{I}\{b_t = \mathcal{F}_{br}(a)\} + \sum_{t=1}^{n(a)}\left(r_{l,t}(a, b_t) - \mu_l(a, \mathcal{F}_{br}(a))\right)\mathbb{I}\{b_t \neq \mathcal{F}_{br}(a)\}\right|$$

$$\leq \left|\sum_{t=1}^{n(a)}\left(r_{l,t}\left(a, \mathcal{F}_{br}(a)\right) - \mu_l\left(a, \mathcal{F}_{br}(a)\right)\right)\mathbb{I}\{b_t = \mathcal{F}_{br}(a)\}\right| + T_{wbr}(a).$$

And when $n(a) - T_{wbr}(a) \geq \mathcal{O}\left(\frac{\log(AB/\delta)}{\varepsilon^2}\right)$, by Lemma 3, with probability at least $1 - \frac{\delta}{ABT}$,

$$\frac{1}{n(a)}\left|\sum_{t=1}^{n(a)} \left(r_{l,t}\left(a, \mathcal{F}_{br}\left(a\right)\right) - \mu_l\left(a, \mathcal{F}_{br}(a)\right)\right)\mathbb{I}\left\{b_t = \mathcal{F}_{br}(a)\right\}\right| \leq \frac{\varepsilon}{8}.$$

So when $n(a) \geq T_{wbr}(a) + \mathcal{O}\left(\frac{\log\frac{AB}{\delta}}{\varepsilon^2}\right)$, and $n(a) \geq \frac{8}{\varepsilon}T_{wbr}(a)$, with probability at least $1 - \frac{\delta}{ABT}$, we have

$$\frac{1}{n(a)}\left|\sum_{t=1}^{n(a)} r_{l,t}(a, b_t) - \mu_l(a, \mathcal{F}_{br}(a))\right| \leq \frac{\varepsilon}{8} + \frac{1}{n(a)}T_{wbr}(a) \leq \frac{\varepsilon}{8} + \frac{\varepsilon}{8} = \frac{\varepsilon}{4}$$

We set $S_0 \geq \mathcal{O}\left(\frac{B}{\varepsilon^3}\log\frac{ABT}{\delta}\right)$, which will guarantee $n(a) \geq \mathcal{O}\left(\frac{B}{\varepsilon^3}\log\frac{ABT}{\delta}\right)$ and the following inequality holds based on above analysis

$$|\hat{\mu}_l(a) - \mu_l(a, \mathcal{F}_{br}(a))| \leq \frac{\varepsilon}{4}.$$

When $n(a) \geq \mathcal{O}\left(\frac{S_0}{\varepsilon^2}\right)$, $\sqrt{\frac{S_0}{n(a)}} \leq \frac{\varepsilon}{8}$, then with probability at least $1 - \frac{\delta}{T}$, for all $a \in \mathcal{A}$, we have,

$$|\text{ucb}_l(a) - \mu_l(a, \mathcal{F}_{br}(a))| \leq \frac{3\varepsilon}{8} < \frac{\varepsilon}{2}.$$

So $\arg\max_{a \in \mathcal{A}} \text{ucb}_l(a) = a_{se}$. Then by the union bound, with probability at least $1 - 3\delta$, we have

$$\mathbb{P}\left[a \neq a_{se}\right] \leq \frac{\delta}{T}.$$

$\square$

# B PROOF FOR SECTION 6

## B.1 PROOF FOR THEOREM 4

**Theorem** (Last iterate convergence of EXP3-FMUCB with noisy side information). *When the leader uses EXP3 with parameter $\alpha = \mathcal{O}\left(T^{-\frac{1}{3}}\right)$, $\eta = \mathcal{O}\left(T^{-\frac{1}{3}}\right)$, the follower uses FMUCB, and let $T_{f,w}$ be the total number of rounds that the follower did not play the best manipulation strategy in $T$ rounds, with probability at least $1 - 3\delta$,*

$$T_{f,w} \le \mathcal{O}\left(\frac{A^2 B}{\alpha \varepsilon^2} \log \frac{ABT}{\delta}\right), and$$

$$\mathbb{P}[a \ne a_{se}] \le \alpha + A \exp\left(-\Delta_3 T^{\frac{2}{3}} + 2\sqrt{2A \log \frac{2}{\delta}} T^{\frac{1}{3}} + \frac{C_2 A^2 B}{\varepsilon^2} \log \frac{ABT}{\delta} \log \frac{1}{\delta}\right),$$

*where $\varepsilon = \min\{\Delta_4, \Delta_5, \Delta_6\}$, $C_2$ is a constant.*

We study the case when the follower fails to do the manipulation when leader plays $a$. We ignore the subscript $t$ when there is no ambiguity, including the following two cases.

**Wrong recognition for wr$(a)$**   We denote the rounds that the follower did not find wr$(a)$ correct for all $a \in \mathcal{A}$ as $t_1$. By Lemma 2, we have

$$t_1 \le \sum_{a \in \mathcal{A}} T_{wwr}(a) \le \mathcal{O}\left(AB \frac{\log(ABT/\delta)}{\Delta_1^2} + AB \log\left(\frac{AB}{\delta}\right)\right).$$

**Wrong recognition of manipulation when wr$(a)$ is correct for all $a \in \mathcal{A}$**   We next study the wrong manipulation rounds when wr$(a)$ is correct for all $a \ne a_{se}$. For any action pair $(a, b) \in \mathcal{A} \times \mathcal{B}$, we consider whether it will be selected as the best manipulation pair or under what condition it will be eliminated by Algorithm 2 when it is not the best manipulation pair. We classify and discuss two different situations:

(i) $\mu_f(a, b) \ge \mu_f(a_{fm}, b_{fm})$, but there exists $\hat{a} = \arg\min_{\hat{a} \ne a} \mu_l(\hat{a}, \text{wr}(\hat{a}))$, $\mu_l(\hat{a}, \text{wr}(\hat{a})) \ge \mu_l(a, b)$. As a reminder, $\mu_l(\hat{a}, \text{wr}(a)) - \mu_l(a, b) = \Delta_6 \ge \varepsilon$. When $n(a, b) > t_2 = \mathcal{O}\left(\frac{\log(ABT/\delta)}{\varepsilon^2}\right)$ and $n(\hat{a}, \text{wr}(\hat{a})) > t_2$, by Lemma 3, the following inequality holds

$$\hat{\mu}_{l,t}(a, b) \le \mu_l(a, b) + \frac{\varepsilon}{8} < \mu_l(\hat{a}, \text{wr}(a)) - \frac{\varepsilon}{4} \le U_{l,t}(\hat{a}, \text{wr}(\hat{a})),$$

which means that $(a, b)$ will be eliminated by Algorithm 2 through Line 4. So it needs at most extra $2t_1$ rounds for action pair $(a, b)$ to eliminate the wrong manipulation pair $(a, b)$.

(ii) $\mu_f(a, b) < \mu_f(a_{fm}, b_{fm})$. When $n(a, b) > t_3 = \mathcal{O}\left(\max\left\{\frac{\log(ABT/\delta)}{\varepsilon^2}, \frac{S_1}{\varepsilon^2}\right\}\right)$, by Lemma 3, we have

$$U_l(a, b) \le \mu(a, b) + \frac{\varepsilon}{4} < \mu(a_{fm}, b_{fm}) < U_l(a_{fm}, b_{fm}).$$

So the extra wrong manipulation rounds for $(a, b)$ is no more than $t_3$.

When the leader plays more than $\mathcal{O}\left(\frac{A \log(1/\delta)}{\alpha} t_0\right)$ rounds, because of the existence of uniform exploration parameter $\alpha$, with probability at least $1 - \delta$, $n(a) \ge t_0$. So by the union bound, with probability at least $1 - 3\delta$,

$$T_{f,w} \le \mathcal{O}\left(\frac{A}{\alpha} \log \frac{1}{\delta} (t_1 + ABt_2 + ABt_3)\right) = \mathcal{O}\left(\frac{A^2 B}{\alpha \varepsilon^2} \log \frac{ABT}{\delta} \log \frac{1}{\delta}\right).$$

We denote $\mu_m(a) = \mu_l(a, \mathcal{F}_{fm}(a))$ in this section when there is no ambiguity. For $a \neq a_{se}$, we have

$$\sum_{t=1}^{T} r_t(a) - r_t(a_{fm}) = \sum_{t=1}^{T} \left( r_t(a) - \mu_m(a) \right) + \left( \mu_m(a) - \mu_m(a_{fm}) \right) + \left( \mu_m(a_{fm}) - r_t(a_{fm}) \right)$$

$$= \sum_{t=1}^{T} \left( r_t(a) - \mu_m(a) \right) - \Delta_3 T + \sum_{t=1}^{T} \left( \mu_m(a_{fm}) - r_t(a_{fm}) \right)$$

$$\leq \mathcal{O} \left( \frac{A \log(1/\delta)}{\alpha} t_1 \right) - \Delta_3 T + 2 s_m B(t_2 + t_3).$$

Analogous to the proof of Theorem 2 in Appendix A.2, we can complete the proof of Theorem 4.

## B.2   PROOF FOR THEOREM 5

**Theorem** (Last iterate convergence of UCBE-FMUCB with noisy side information)**.** *When the leader uses UCBE with* $S_0 = \mathcal{O}\left(\frac{B}{\varepsilon^3}\log\frac{ABT}{\delta}\right)$ *and* $\varepsilon = \min\{\Delta_4, \Delta_5, \Delta_6\}$*, the follower uses FMUCB, and let* $T_{f,w}$ *be the total number of rounds that the follower did not play the best manipulation strategy in* $T$ *rounds, with probability at least* $1 - 3\delta$*,*

$$T_{f,w} \leq \mathcal{O}\left(\frac{AB}{\varepsilon^2}\log\frac{ABT}{\delta}\right),$$

*and for* $T \geq \mathcal{O}\left(\frac{AS_0}{\Delta_3^2}\right)$*,* $\mathbb{P}\left[a_T \neq a_{fm}\right] \leq \frac{\delta}{T}$*.*

Since the leader who uses UCBE will do pure exploration for each $a \in \mathcal{A}$ with $\mathcal{O}\left(\frac{B}{\varepsilon^3}\log\frac{ABT}{\delta}\right)$ according to the analysis in Appendix A.3, where $\varepsilon = \min\{\Delta_4, \Delta_5, \Delta_6\}$. So by the union bound, with probability at least $1 - 3\delta$,

$$T_{f,w} \leq \mathcal{O}\left(t_1 + ABt_2 + ABt_3\right) = \mathcal{O}\left(\frac{AB}{\varepsilon^2}\log\frac{ABT}{\delta}\right).$$

Similar to the proof of Theorem 3, we bound the difference between the accumulative reward the leader receives and the accumulative reward under best manipulation strategy,

$$\left|\sum_{t=1}^{n(a)} r_{l,t}(a, b_t) - \mu_l(a, \mathcal{F}_{fm}(a))\right|$$

$$\leq \left|\sum_{t=1}^{n(a)} \left(r_{l,t}\left(a, \mathcal{F}_{fm}(a)\right) - \mu_l\left(a, \mathcal{F}_{fm}(a)\right)\right) \mathbb{I}\left\{b_t = \mathcal{F}_{fm}(a)\right\}\right| + \sum_{t=1}^{n(a)} \mathbb{I}\left\{b_t \neq \mathcal{F}_{fm}(a)\right\}.$$

And based on the proof of Theorem 4 in Appendix A.3 and Lemma 2, we have

$$\sum_{t=1}^{n(a)} \mathbb{I}\left\{b_t \neq \mathcal{F}_{fm}(a)\right\} \leq T_{wwr}(a) + Bt_2 + Bt_3 = \mathcal{O}\left(\frac{B}{\varepsilon^2}\log\frac{ABT}{\delta}\right).$$

Analogous to the proof of Theorem 3 in Appendix A.3, we can complete the proof of Theorem 5.

## C FOLLOWER'S PESSIMISTIC MANIPULATION

We denote $\mathbf{Q}(\mathcal{F}) = \arg\max_{a \in \mathcal{A}} \mu_l(a, \mathcal{F}(a))$. Because there may be multiple actions in $\mathcal{A}$ with the same largest $\mu_l(a, \mathcal{F}(a))$, so there may be more than one element in $\mathbf{Q}(\mathcal{F})$. From the follower's perspective, pessimistically, the leader will break the tie using $a = \arg\min_{a \in \mathbf{Q}(\mathcal{F})} \mu_f(a, \mathcal{F}(a))$. Assume pessimistically tie-breaking is more reasonable considering that the leader uses no-regret learning algorithm, since the leader who uses no-regret learning algorithm may not play one action in $\mathbf{Q}(\mathcal{F})$ stably. So it is important for the follower to choose an appropriate manipulation strategy that will maximize its own reward in the long run taking into account the leader's tendency. When the follower makes the assumption that the leader will break the tie pessimistically, the optimization problem of the follower is

$$\max_{\mathcal{F}} \min_{(a',b') \in \mathcal{A} \times \mathcal{B}} \mu_f(a', b')$$
$$\text{s.t.} \quad a' \in \arg\max_{a \in \mathcal{A}} \mu_l(a, \mathcal{F}(a)) \tag{12}$$
$$b' = \mathcal{F}(a')$$

We present the Algorithm 3 for solving optimization problem (12), which is similar to Algorithm 1.

---

**Algorithm 3:** Follower's best manipulation towards a pessimistic leader

---

**Input:** Candidate set $\mathcal{K} = \mathcal{A} \times \mathcal{B}$, $\mathbf{H} = \{\}$, $\mu_l(\cdot, \cdot)$, $\mu_f(\cdot, \cdot)$
1: Candidate manipulation pair $(a', b') = \arg\max_{(a,b) \in \mathcal{K}} \mu_f(a, b)$
2: **if** $\mu_f(a_p, b_p) < \mu_f(a', b')$ **then**
3:     $\mathcal{F} = \{\mathcal{F}(a') = b'\} \cup \{\mathcal{F}(a) = \arg\min_{b \in \mathcal{B}} \mu_l(a, b) : a \neq a'\}$
4:     Let $\mathbf{Q} = \arg\max \mu_l(a, \mathcal{F}(a))$
5:     **if** $(a', b') \in \mathbf{Q}$ **then**
6:         Denote the current $\mathcal{F}$ as $\mathcal{F}_{a_p, b_p}$, $\mathbf{H} \cup \arg\min_{(a,b) \in \mathbf{Q}} \mu_f(a, b)$, $(a_p, b_p) = \arg\max_{(a,b) \in \mathbf{H}} \mu_f(a, b)$
7:     **end if**
8:     Eliminate $(a, b)$ from candidate set: $\mathcal{K} \leftarrow \mathcal{K} \backslash (a', b')$ and return to Line 1
9: **else**
10:     $\mathcal{F}_{opt} = \mathcal{F}_{a_p, b_p}$
11: **end if**
**Output:** The response function $\mathcal{F}_{opt}$

---