# OpenReview forum: "Decentralized Online Learning in General-Sum Stackelberg Games"
_auai.org/UAI/2024/Conference — UAI 2024 poster_

### Official Review · Reviewer_oVhw · 2024-03-01

**Q2-1 Originality-Novelty:** 3
**Q2-2 Correctness-Technical Quality:** 3
**Q2-5 Clarity Of Writing:** 3

**Q1 Summary And Contributions:**

This paper considers the problem of decentralized online learning in general-sum Stackelberg games (leader-follower games). This paper investigates two setups: (1) the limited information setup, where the follower can only obtain his/her own reward information in each round; and (2) the side information setup, where the follower can also obtain the information of the leader in each round. In the side information setup, this paper further studies two setups: (a) the omniscient follower setup (like full information setup in online learning), where the follower can know the complete knowledge of the reward function; and (b) the noisy side information setup (like bandit setup in online learning), where the follower can only attain noisy bandit feedbacks of the himself/herself and the leader. In the limited information setup, the authors design two kinds of algorithms (Exp3-UCB and UCBE-UCB) and obtain last-iterate convergence result. In the side information setup with an omniscient follower, the authors proposed a manipulation algorithm for the follower. And in the noisy side information setup, the authors proposed a UCB-type variant of the FBM algorithm for estimation. This algorithm, along with the leader running Exp3 or UCBE as the limited information setup, also enjoys similar last-iterate guarantees as in the limited information setup. Finally, empirical studies validate the effectiveness of the proposed methods.

**Q2-3 Extent To Which Claims Are Supported By Evidence:**

3: Good: the main claims are supported by convincing evidence (in the form of adequate experimental evaluation, proofs, (pseudo-)code, references, assumptions).

**Q2-4 Reproducibility:**

3: Good: key resources (e.g. proofs, code, data) are available and key details (e.g. proofs, experimental setup) are sufficiently well-described for competent researchers to confidently reproduce the main results.

**Q3 Main Strengths:**

This paper focuses on the classic problem of Stackelberg games, which can find many applications in real-world scenarios. This paper studies a valuable variant of it by focusing on the decentralized version. The presentation is clear, intuitive, and thus easy to follow for new readers. The descriptions of the algorithms are also detailed, and clear. The last-iterate convergence results are meaningful and appealing. The empirical studies are sufficient.

**Q4 Main Weakness:**

I do not see major weaknesses in this paper. However, I have a question about Theorem 2 (analogously, Theorem 4), which may be listed in Q5.

**Q5 Detailed Comments To The Authors:**

I understand that Theorem 3 is a good result since we usually choose $\delta = O(1/T)$ to ensure a high probability bound. In this case, the probability that $a_T \ne a_{se}$ is less than $O(1/T^2)$, which is favorable enough. However, I do not understand how good the result in Theorem 2 is. The probability that $a_T \ne a_{se}$ is described by a complicated formulation. I guess this result is not as good as that in Theorem 2? Can the authors provide some explanations on how good this result is compared to Theorem 3? Also, a discussion of the result should be provided below Theorem 2.

**Q9 Complying With Reviewing Instructions:**

Yes

---

> ### Author Rebuttal · Authors · 2024-04-04
>
> >Can the authors provide some explanations on how good this result is compared to Theorem 3?
>
> The formulations of the bounds presented in both Theorem 2 and Theorem 3 are conventional within the learning theory community, attributable to divergent analytical approaches applied to EXP3 and UCB type algorithms.
> Generally speaking, Theorem 2 introduces a decay rate of $\mathcal{O}(T^{-1/3} + \exp(-T^{2/3}))$, contrasting with the linear decay in $T$ observed in Theorem 3. This distinction suggests a comparative advantage for Theorem 3, contingent upon a specific constraint on $T$. Nevertheless, given that both bounds are influenced by the gaps, denoted as $\Delta_1, \Delta_2$, a thorough comparison necessitates consideration of these gap values. Thanks for pointing this out. We will add this discussion in our revision.

---

### Official Review · Reviewer_E8aq · 2024-03-22

**Q2-1 Originality-Novelty:** 3
**Q2-2 Correctness-Technical Quality:** 3
**Q2-5 Clarity Of Writing:** 3

**Q1 Summary And Contributions:**

This paper studies an online learning problem in general-sum Stackelberg game. The authors consider two settings corresponding to different levels of information received by the  follower. In the limited information setting, in which the follower only observes the reward of its own, the authors prove the convergence of general-sum Stackelberg equilibrium when both players use specific no-regret learning algorithms. In the side-information setting,  in which the follower has extra side information on the leader’s reward as well,  the authors distinguish full and exact information and noisy side information, where the follower needs to learn the leader’s reward information from noisy bandit feedback in the online process. In both cases, they provide algorithms and give regret and last step iterate guarantees.

**Q2-3 Extent To Which Claims Are Supported By Evidence:**

3: Good: the main claims are supported by convincing evidence (in the form of adequate experimental evaluation, proofs, (pseudo-)code, references, assumptions).

**Q2-4 Reproducibility:**

3: Good: key resources (e.g. proofs, code, data) are available and key details (e.g. proofs, experimental setup) are sufficiently well-described for competent researchers to confidently reproduce the main results.

**Q3 Main Strengths:**

This paper is well-written and clear. The problem is obviously relevant, since Stackelberg games find applications in taxation, auctions etc. The theoretical results prove that there is an advantage in manipulating the the leader in the side information setting.

**Q4 Main Weakness:**

Some choices are not clearly justified. Why is the last step iterate convergence the right performance indicator?
Why would UCB-E or EXP3 be used by the leader? What are the comparative advantages of both of them?

**Q5 Detailed Comments To The Authors:**

Please refer to the comments above

**Q9 Complying With Reviewing Instructions:**

Yes

---

> ### Author Rebuttal · Authors · 2024-04-04
>
> > Why is the last step iterate convergence the right performance indicator?
>
> Last-iterate convergence refers to the phenomenon where the sequence of actions converges to the optimal action or the sequence of policies converges to the optimal policy. It represents a particularly strong form of convergence, which is inevitably a stronger notion than the average-iterate convergence [Wu et. al. 2022, Cai et. al. 2024, Abe et. al. 2023]: an algorithm demonstrating last-iterate convergence is proven to also be capable of achieving average-iterate convergence whereas the reverse is not necessarily true.
>
> > Why would UCB-E or EXP3 be used by the leader? What are the comparative advantages of both of them?
>
> Both algorithms, UCB-E and EXP3, are chosen for their ability to ensure sufficient exploration, which is crucial for guaranteeing convergence to the Stackelberg equilibrium as highlighted in Section 4. Theorem 1 emphasizes the critical notion that algorithms being deficient in exploration may not successfully converge to Stackelberg equilibrium. The combination of both algorithms with the follower's UCB algorithm ensures convergence to the Stackelberg equilibrium.

---

### Official Review · Reviewer_e44x · 2024-03-23

**Q2-1 Originality-Novelty:** 3
**Q2-2 Correctness-Technical Quality:** 3
**Q2-5 Clarity Of Writing:** 3

**Q1 Summary And Contributions:**

The paper focuses on an online learning problem in general-sum Stackelberg games where players act in a decentralized and strategic manner. The study examines two settings based on the type of information available to the follower: a limited information setting where the follower only observes its own reward, and a side information setting where the follower has extra information about the leader's reward. The key contributions of this paper include:

Demonstrating that in scenarios where followers have limited information, responding optimally to the leader's actions is the best strategy. However, with side information, followers can strategically manipulate signals to sway the leader's strategy towards more favorable outcomes for themselves.
Introducing decentralized online learning algorithms for both players in these settings and deriving last iterate convergence and sample complexity results for these algorithms. Notably, a new manipulation strategy is designed for the follower in the side information setting, showing an intrinsic advantage over the best response strategy.
Empirical validation of the theoretical findings, demonstrating the effectiveness of the proposed learning algorithms and strategies in achieving convergence and highlighting the advantage of manipulation strategies for the follower in the side information setting​​.

**Q2-3 Extent To Which Claims Are Supported By Evidence:**

3: Good: the main claims are supported by convincing evidence (in the form of adequate experimental evaluation, proofs, (pseudo-)code, references, assumptions).

**Q2-4 Reproducibility:**

3: Good: key resources (e.g. proofs, code, data) are available and key details (e.g. proofs, experimental setup) are sufficiently well-described for competent researchers to confidently reproduce the main results.

**Q3 Main Strengths:**

1. A key strength of the paper is its extensive theoretical support for the authors' claims, which includes not just last iterate convergence but also sample complexity results.
2. Another strength is the inclusion of empirical results that corroborate the theoretical findings. The blend of theoretical rigor with experimental validation provides a holistic view of the research's validity.

**Q4 Main Weakness:**

1. The study's focus on normal-form general-sum games with a Stackelberg-like sequential setup might be perceived as less complex compared to environments with constraints and continuous actions, as discussed in [Goktas+ 2022]. This simplification may limit the paper's appeal in addressing or being applicable to more intricate or real-world strategic interactions.
2. The research effectively demonstrates the algorithms' strengths in scenarios where multiple equilibria exist, underscoring their utility in such contexts. However, a weakness is the limited experimental exploration in other settings, such as those with unique equilibria.

**Q5 Detailed Comments To The Authors:**

Comment:
1. The application of terms like "last-iterate convergence" (LIC) to algorithms like FBM and FMUCB necessitates careful consideration. These algorithms diverge from those traditionally associated with LIC  [Mertikopoulos et al., 2018, Daskalakis and Panageas, 2018], suggesting a need for a discussion on their categorization and implications.
2.  An expanded review of related research would significantly strengthen your paper, especially regarding LIC in bandit and noisy feedback environments. Consider incorporating the following studies to provide a richer context for your contributions:

[Heliou+ NeurIPS 2017, "Learning with Bandit Feedback in Potential Games"]

[Abe+ AISTATS 2023, "Last-Iterate Convergence with Full and Noisy Feedback in Two-Player Zero-Sum Games"]

[Cai+ NeurIPS 2023, "Uncoupled and Convergent Learning in Two-Player Zero-Sum Markov Games with Bandit Feedback"]

Question:
1. Could you provide examples of real-world problems that can be modeled with the problem settings used in your study?
2. In the context of general-sum games, equilibrium selection is a significant challenge. Considering this, can we always regard algorithms that achieve convergence to specific equilibria as effective?

**Q9 Complying With Reviewing Instructions:**

Yes

---

> ### Author Rebuttal · Authors · 2024-04-04
>
> > ... might be perceived as less complex compared to environments with constraints and continuous actions, as discussed in [Goktas+ 2022].
>
> The research conducted by [Goktas+ 2022] primarily approaches the problem from an optimization viewpoint, focusing on the full comprehension of utility functions and examining the existence and solvability of specific equilibria in stochastic zero-sum games. Our study, however, adopts a **learning** perspective, emphasizing the necessity for players to learn from noisy bandit feedback in an online setting. We aim to design practical algorithms accompanied by convergence analysis, a framework that is similarly employed in related works by Bai et al. (2021) and Kao et al. (2022), which also demonstrate significant real-world applicability.
>
> Moreover, our paper introduces the novel challenge of decentralized learning and manipulation in general-sum games based on noisy bandit feedback. Even within this "simpler" setting, we uncover non-trivial and compelling findings, including convergence to Stackelberg equilibrium and the development of a learnable best manipulation strategy for the follower. This highlights the distinctive contributions and insights our study offers to the field. We do think that the more complex settings like in [Goktas+ 2022] are worth further exploration in the future.
>
> > Could you provide examples of real-world problems that can be modeled with the problem settings used in your study?
>
> Our study's framework finds relevance in a variety of real-world scenarios, including:
> 1. Taxation and AI Economist [Bai et al. (2021)]: This model involves a government (leader) setting tax rates, and citizens (followers) deciding on their labor contributions based on these rates.
> 2. Repeated Auctions with a Buyer[Amin et al., 2013]: This setting involves repeated auctions with a strategic buyer.
> 3. Seller-Buyer Problem: This common economic model involves a seller setting prices without prior knowledge of the buyer’s willingness or ability to purchase, embodying a classic Stackelberg game dynamic.
>
> All these examples follow our game structure and show the broad applicability of our study's problem settings, encompassing economics and market dynamics.
>
> > A weakness is the limited experimental exploration in other settings, such as those with unique equilibria.
>
> It appears there may be a typo or misunderstanding; our study does encompass scenarios with unique equilibria. Our core and primary objective is to explore the basic but most fundamental problem, so we adopt certain assumptions for simplicity, clarity. Our assumptions can be adjusted to cover a wider array of situations. For example:
>
> 1. In Section 4, we address the situation of multiple Stackelberg equilibria by extending our theorem to include an average convergence result. This implies that over time, the expected average reward for the leader aligns with the reward at the Stackelberg Equilibrium, expressed as $\frac{1}{T}\sum_{t=1}^T \mu_l(a_t, b_t) \rightarrow \mu_l(a_{se}, b_{se}$, thus covering scenarios with multiple equilibria.
> 2. For Section 6, we introduce a variant of the FMUCB algorithm for the follower, specifically designed for scenarios where multiple Stackelberg equilibria exist. This is detailed further in Appendix C.
>
> >In the context of general-sum games, equilibrium selection is a significant challenge. Considering this, can we always regard algorithms that achieve convergence to specific equilibria as effective?
>
> In certain general-sum games, e.g. Nash equilibria and Correlated equilibria, equilibrium selection is indeed a challenge. But in the context of our work on Stackelberg games, equilibrium selection does not present a problem. The leader, having the advantage of moving first, can choose among multiple Stackelberg equilibria without affecting its reward; each choice yields the same benefit. Therefore, in our study, algorithms that converge to specific Stackelberg equilibria are deemed effective.
>
> > The application of terms like LIC necessitates careful consideration.
>
> Last-iterate convergence refers to the phenomenon where the sequence of actions converges to the optimal action or the sequence of policies converges to the optimal policy, formulated as $\lim_{t \to \infty} a_t = a_\star$ or $\lim_{t \to \infty} \pi_t = \pi_\star$. The last-iterate convergence results from our work, as well as the related works you mentioned, align with this definition. For example, in the findings of Mertikopoulos et al., 2018 (Theorem 4.1), the monotonic decline of $D(x^*, X_n)$ decreases monotonically to 0 suggests $\lim_{n\to\infty} X_n = x^*$. Similarly, Abe et al., 2023 (Theorem 5.1) deduce that $\lim_{t\to\infty} \pi^t = \pi^{\mu, r}$, and Cai et. al. 2024's upper bound for duality gap of $(x_T, y_T)$ implies $\lim_{T\to\infty} (x_T, y_T) = (x_*, y_*)$. Our findings corroborate this view, indicating that $\lim_{t \to \infty} P(a_t \neq a_\star)=0$ leads to $\lim_{t \to \infty} a_t = a_\star$.

---

### Official Review · Reviewer_p72W · 2024-03-23

**Q2-1 Originality-Novelty:** 2
**Q2-2 Correctness-Technical Quality:** 2
**Q2-5 Clarity Of Writing:** 2

**Q10 Ethical Concerns:**

The submission does not raise potential ethical concerns.

**Q1 Summary And Contributions:**

This paper studies no-regret learning for the repeated play of general-sum Stackelberg games when agents do not know their payoff functions. The game has two agents: leader and follower. Agents have finitely many actions. They receive bounded (possibly noisy) payoffs.
The repeated play scheme is as follows:
- (1) The leader plays an action.
- (2) The follower observes the leader's action and makes a response
- (3) The leader cannot observe the follower's action but only receives a noisy reward
- (4) The follower either receives her own (noisy) reward or her and the leader's (noisy) rewards.

Here, (3) and (4) play important roles in the conclusions drawn. Particularly due to (3), no-regret learning dynamics (such as UCB vs UCB) do not necessarily reach equilibrium (Theorem 1). The authors showed that EXP3 (with extra exploration) vs UCB can reach equilibrium. Furthermore, when the follower has access to the leader's payoff, the follower can manipulate the leader to outcomes preferable for the follower.

**Q2-3 Extent To Which Claims Are Supported By Evidence:**

2: Fair: the main claims are somewhat supported by evidence (but the experimental evaluation may be weak, or does not match entirely with the claims, important baselines may be missing, proofs contain important ideas but lack rigor, algorithmic details are only discussed superficially, references are imprecise, assumptions are not sufficiently motivated or explicated, etc.).

**Q2-4 Reproducibility:**

3: Good: key resources (e.g. proofs, code, data) are available and key details (e.g. proofs, experimental setup) are sufficiently well-described for competent researchers to confidently reproduce the main results.

**Q3 Main Strengths:**

The paper focuses on general-sum games.
The paper provides last-iterate convergence guarantees.

**Q4 Main Weakness:**

Section 3:
- The formulation of the Stackelberg game is confusing. Although the introduction discusses commitment by the leader, Section 3 does not discuss it.
- The repeated game scheme describes the repeated play of a two-stage sequential game rather than a Stackelberg game.

Section 4:
- The follower faces stochastic bandit problems specific to the leader's actions since she can observe the leader's actions. Therefore, she can follow UCB-type algorithms. On the other hand, the leader does not observe the follower's actions. Therefore, she does not face a (stationary) stochastic bandit problem since the payoffs received also depend on the follower's (unobserved) actions. In other words, contrary to the noise, the follower's actions are not necessarily stationary. Indeed, if the leader plays an action frequently, then the follower's response becomes more and more stationary as the follower can solve the associated stationary stochastic bandit problems she faces. Therefore, further exploration can resolve this issue. My point is that we would not have non-convergence results if the leader were observing the follower's actions. This should have been emphasized.

Section 5:
- The paper states, "When the follower uses the response set F, the leader will take actions that maximize its own reward given F". This describes a Stackelberg game where the paper's follower is the leader committing to strategy F, and the paper's leader is the follower reacting to the strategy by knowing it". Note that there can be Stackelberg games where the follower takes action first by knowing the leader's committed response strategy. Therefore, the conclusion about the paper's follower misleading the paper's leader is not counter-intuitive and indeed expected due to the commitment power of the leader (the paper's follower) in the Stackelberg games. The bandit setting does not pose many challenges as the follower (the paper's leader) faces a stochastic bandit problem, as in Section 4. However, different from Section 4, the leader (the paper's follower) now knows the payoffs.

**Q5 Detailed Comments To The Authors:**

Could the authors address my concerns stated in Q4?

**Q9 Complying With Reviewing Instructions:**

Yes

---

> ### Author Rebuttal · Authors · 2024-04-04
>
> > The formulation of the Stackelberg game is confusing. Although the introduction discusses commitment by the leader, Section 3 does not discuss it.
>
> Our formulation is explicitly defined in Section 3 Preliminary -- We do not investigate on a commitment leader. The study on a commitment leader that we mentioned in the Introduction and Related Work Sections are from related works but not from our setting.
>
> > The repeated game scheme describes the repeated play of a two-stage sequential game rather than a Stackelberg game.
>
> The formulation of our fucus is indeed repeated Stackelberg game, which from our understanding is equivalent to a repeated play of a two-stage sequential game as you say. Our detailed definition of the repeated Stackelberg game is in the Section 3 and the formulation is clear. This setting is also widely used by related works [Haghtalab et al. (2022), Bai et al. (2021) and Kao et al. (2022).] Could you please help us clarify if our definition makes sense?
>
>
> > My point is that we would not have non-convergence results if the leader were observing the follower's actions.
>
> In fact, the proof of Theorem 1 (as well as all the other theorems in the paper) does not rely on the assumption that the leader can not observe the follower's action. Even if the leader can observe the follower’s action, the no-regret learning dynamics (such as UCB-UCB) still do not necessarily reach an equilibrium.  The non-convergence of Theorem 1 is mainly due to the lack of exploration: the follower does not find the best response, so the leader can not learn the Stackelberg equilibrium. All our results do not rely on this assumption. Our original consideration assuming the leader not observing the follower's actions was because we thought it was a more practical setting as in many real-world cases the leader can not observe the follower's actions. Thanks for pointing this out. We will remove this assumption in the revised version.
>
> > The bandit setting does not pose many challenges as the follower (the paper's leader) faces a stochastic bandit problem, as in Section 4. However, different from Section 4, the leader (the paper's follower) now knows the payoffs.
>
> Section 5 serves as a preliminary exploration in preparation for the more complex discussions in Section 6. In Section 5, we investigate the best manipulation strategy under the assumption that the follower is omniscient. This sets the stage for Section 6, where we delve into the more intricate setting of learning the best
> manipulation strategy with noisy side information. The technical challenges confronted in Section 6 can be attributed to several factors: 1) the hierarchical problem structure, 2) the general-sum game structure, 3) the noisy bandit feedback, and 4) the practical constraints of decentralized learning. Section 6 aims to unravel the process through which the follower can learn the most effective manipulation strategy within this dynamic and multifaceted environment. The confluence of these elements distinctly marks the scenario as more complex than a straightforward stochastic bandits problem.

---

### Official Review · Reviewer_TEFy · 2024-03-25

**Q2-1 Originality-Novelty:** 3
**Q2-2 Correctness-Technical Quality:** 3
**Q2-5 Clarity Of Writing:** 3

**Q1 Summary And Contributions:**

This paper considers the problem of finding Stackelberg equilibrium in online learning settings. In limited information setting, the follower's best strategy is to learn best response against each of leader's strategy, which becomes a stochastic bandit problem. The author proved results that if both players adopted some combination of bandit algorithms then the profile will converge to Stackelberg. In full-information setting where the follower can observe the leader's reward, follower has the power of manipulating leader's decision, assuming it is using a no-regret algorithm, to gain more benefit in long run. An interesting maniuplation scheme was devised where the follower response by trying to minimze the leader's payoff while maximizing its own payoff. Theoretical results were given for both exact feedback and nosiy feedbacks.

**Q2-3 Extent To Which Claims Are Supported By Evidence:**

2: Fair: the main claims are somewhat supported by evidence (but the experimental evaluation may be weak, or does not match entirely with the claims, important baselines may be missing, proofs contain important ideas but lack rigor, algorithmic details are only discussed superficially, references are imprecise, assumptions are not sufficiently motivated or explicated, etc.).

**Q2-4 Reproducibility:**

3: Good: key resources (e.g. proofs, code, data) are available and key details (e.g. proofs, experimental setup) are sufficiently well-described for competent researchers to confidently reproduce the main results.

**Q3 Main Strengths:**

Problem are well-motivated. The manipulation scheme in algorithm 1 provides interesting insights.

**Q4 Main Weakness:**

Some assumptions are not reasonable and some details are missing.

**Q5 Detailed Comments To The Authors:**

Generally I think this paper studies a very important problem and give an interesting solution. However there are several points I am not very clear:

1. Why the leader's strategy has to be no-regret learning based? In principle, leader can choose any dynamical algorithm mapping from history to actions. Also, why do you only consider pure strategy Stackelberg? I think it is more common to consider the leader is committing to a mixed strategy.

2. I think the assumption that the game has a unique Stackelberg equilibrium and a unique BR for every leader's action is very limited and ignore many interesting cases when you need to choose between Strong SE and Weak SE.

3. In noisy feedback settings, what if the leader never choose a particular strategy in the history?

**Q9 Complying With Reviewing Instructions:**

Yes

---

> ### Author Rebuttal · Authors · 2024-04-04
>
> > Why must the leader's strategy be based on no-regret learning?
>
> No-regret learning algorithms are both simple and effective within the context of game dynamics and are commonly adopted in similar game setting [Anagnostides et. al. 2022, Daskalakis et. al. 2021,  Lauffer et. al.2022]. In the Stackelberg game under consideration, the leader aims to identify and implement the optimal action (denoted as $a_{\text{se}}$) in hindsight. In order to let the sequential action pairs to converge towards the Stackelberg equilibrium, the leader must at least achieve a state of no regret. Arbitrary algorithm mapping from history to actions may lead to non-convergence (Theorem 1 gives an example).
>
> > Why do you only consider pure strategy Stackelberg equilibria?
>
> Our focused setting of a pure strategy follows that of Bai et al. (2021) and Kao et al. (2022). Beside this, The focus of this paper is the new challenge of decentralized learning or manipulating in general-sum games from noisy bandit feedback, which we find already exhibiting some non-trivial and interesting results (convergence to Stackelberg equilibrium and learnable best manipulation strategy for the follower) even in the pure strategy setting. By concentrating on pure strategies, we aim to establish a clear and solid foundation from which to explore the dynamics of Stackelberg games.
>
> Indeed, mixed strategy makes more sense for one-shot games, and some specific game classes such as Stackelberg security games. But in many cases, the pure strategy is indeed observable, especially in the online bandit games. Sometimes it is hard for the follower to see the committed mixed strategy of leaders, and it is hard to define the noisy bandit feedback when follower responds to a mixed strategy in many practical online scenarios (receiving an expected noisy feedback is not a practical assumption in many real-world cases). We do agree mixed strategy is interesting and practical which is definitely worth further investigation.
>
> > The assumption of a unique Stackelberg equilibrium and a unique Best Response (BR) for every leader's action seems very limiting.
>
> [TL;DR] Like what we mentioned above, our primary focus is on understanding the overarching process of converging to an equilibrium in a decentralized manner, so we adopt certain assumptions for simplicity and clarity. This is the basic version but also the most fundamental problem.
>
> The assumptions mentioned can indeed be relaxed to accommodate a broader range of scenarios. Specifically, in cases where the follower's best response is not singular, one viable approach is to implement a tie-breaking rule. This rule would allow the follower to play from a set of best responses according to a predefined distribution. Particularly in an online setting, it would be reasonable for the leader to choose an action that maximizes their own reward under the assumption that the follower, employing a Uniform Confidence Bound (UCB) algorithm, plays all best responses with equal probability.
>
> For scenarios presenting multiple Stackelberg equilibria, our theorem is adaptable to provide an average convergence result, symbolized as $\lim_{t \to \infty} \frac{1}{T} \mathbb{E}[\sum_{t=1}^T \mu_l(a_t) - \max_{a\in\mathcal{A}}\mu_l(a)] = 0$. This implies the leader's expected reward gradually aligns with the Stackelberg Equilibrium reward, thus accommodating the presence of multiple equilibria.
>
> However, these specifics are not the main focus of our investigation. Our core and primary research objective is to explore the overarching dynamics of strategic behavior in a repeated, decentralized, general-sum Stackelberg game.
>
>
> > In noisy feedback settings, what if the leader never choose a particular strategy (action) in the history?
>
> First of all, our study is based on a tabular setting, so we do not consider generalizing to unseen actions by approximation. In a tabular setting, if the leader consistently omits a specific strategy from their selection history, we adjust the leader's action set by excluding this action, denoting the modified set as $\{\mathcal{A}^\prime = \mathcal{A} \setminus \{a\}\}$. Consequently, the game is redefined to a new game $\{\mathcal{A}^\prime, \mathcal{B}, \mu_l, \mu_f\}$.
>
> - In Section 4, as we said, lack of exploration may lead to non-convergence of Stackelberg equilibrium. If the leader never chooses a particular strategy in history then the leader may miss the best action and provide no guarantee of convergence to the SE of the original game.
>
> - In Section 6, if the leader never chooses a particular strategy in history then the follower will find best manipulation strategy for the redifined new game.
>
> In our work, when the leader uses EXP3 or UCBE, the leader intrinsically does the exploration and the case “the leader never chooses a particular strategy in history” will not happen.

---

### Meta-Review · Area_Chair_GKZF · 2024-04-15

There is a consensus among the reviewers recommending a weak acceptance (except for one of the reviewers who recommends borderline rejection). Overall, the tone of all the reviewers is positive about the technical contribution of the paper, although they raise some concerns about the underlying assumptions.